# Hyperdimensional Probe: Decoding LLM Representations via Vector Symbolic Architectures

## Abstract

Despite their capabilities, Large Language Models (LLMs) remain opaque with limited understanding of their internal representations. Current interpretability methods either focus on input-oriented feature extraction, such as supervised probes and Sparse Autoencoders (SAEs), or on output distribution inspection, such as logit-oriented approaches. A full understanding of LLM vector spaces, however, requires integrating both perspectives, something existing approaches struggle with due to constraints on latent feature definitions. We introduce the *Hyperdimensional Probe*, a hybrid supervised probe that combines symbolic representations with neural probing. Leveraging Vector Symbolic Architectures (VSAs) and hypervector algebra, it unifies prior methods: the top-down interpretability of supervised probes, SAE's sparsity-driven proxy space, and output-oriented logit investigation. By combining the supervised learning paradigm of traditional probes with the dictionary-based representation principle of SAEs, our approach enables deeper input-focused feature extraction while supporting output-oriented analysis. Our experiments demonstrate that our approach consistently extracts meaningful semantic information across different LLMs, embedding sizes, and configurations, uncovering concept-oriented insights into LLM inference across two distinct scenarios: input-completion tasks and QA-focused text generation. VSA-based probing overcomes the limitations of logit-based analyses, which are constrained by the model's token vocabulary, while also mitigating the noisier interpretability outcomes often produced by SAEs in settings with a bounded conceptual feature space. By supporting a joint investigation of input-output features, this work advances the semantic understanding of neural representations while unifying the complementary perspectives of prior methods.

## 1 Introduction

The black-box nature of LLMs restricts the interpretability of their internal representations, motivating efforts to extract human-interpretable features from their vector spaces (Park et al., 2023). Such efforts typically fall into two probing perspectives (Ghandeharioun et al., 2024): *Output Distribution Inspection*, and *Feature Extraction*. These serve different yet complementary goals: feature-extraction methods target latent input features, whereas output-oriented inspection examines misalignment between a model's internal state and its textual outputs, a task also known as Eliciting Latent Knowledge (ELK). Comprehensive interpretability of LLM vector spaces would however require combining both, revealing high-level input features that may underlie such misalignment.

Probing approaches for output distribution inspection, such as lens-style (Nostalgebraist, 2020; Belrose et al., 2023) and patching-based methods (Ghandeharioun et al., 2024), project the model's embedding space into its output vocabulary space, *limiting the inspection to layer-wise projected next-token predictions* in ELK-centered, output-constrained settings. In contrast, input-oriented feature extraction is currently performed using Sparse Autoencoders (SAEs), and classification-oriented supervised probes. The supervised probes map the embedding space into latent features to assess how much information about them is encoded using a top-down probing strategy (Gurnee & Tegmark, 2023; Marks & Tegmark, 2023; Diego Simon et al., 2024). This mapping paradigm faces two key challenges, as its probes are: *tailored to task-specific features*, limiting generalizability; and *trained directly on the final features*, making it challenging to distinguish information

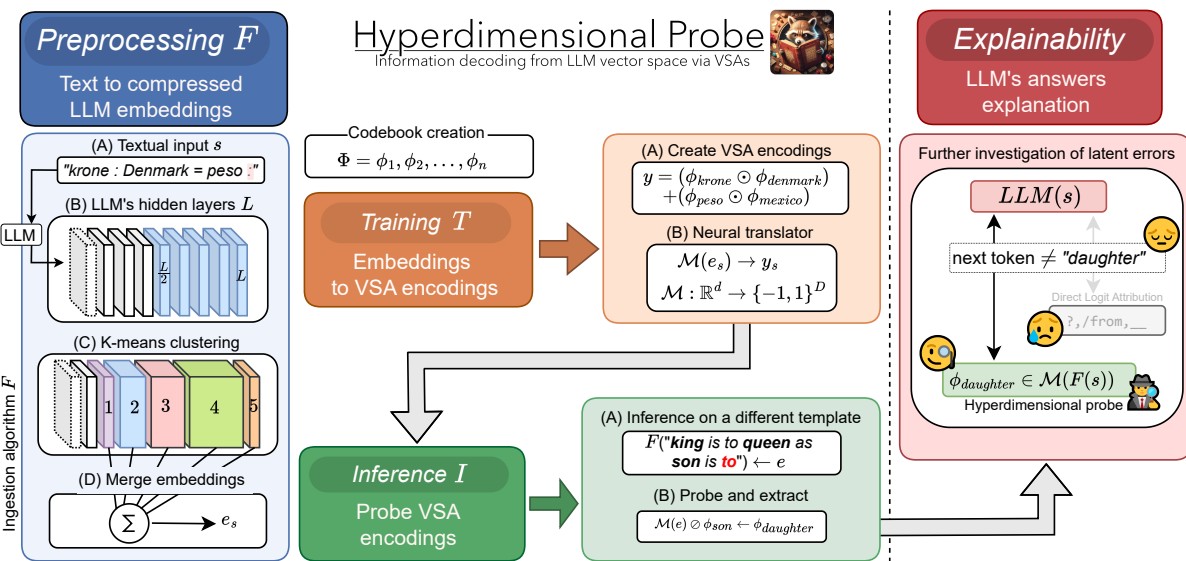

Figure 1: We first compress model's internal state using dimensionality-reduction steps on LLM embeddings ($F$, **blue**). Next, we train a neural VSA encoder to map these compressed neural embeddings into a bounded proxy space: VSA encodings ($T$, **orange**). We then perform concept extraction by querying this proxy space using hypervector algebra ($I$, **green**). Lastly, extracted concepts ultimately enable deeper analysis on model's internal state and its outputs (**red**).

decoding from probe learning. On the other hand, SAEs provide general-purpose feature extraction via a bottom-up probing strategy on a layer-localized proxy space. This, however, reveals an *unbounded feature space* that requires *naming* and *filtering* for interpretability. Meanwhile, comprehensive interpretability of neural embeddings would require integrating *input-driven feature extraction with output-oriented analysis* to identify the internally represented input–output features during LLM inference.

Our work focuses on *input-oriented feature extraction for output-oriented analysis*. Specifically, we investigate how semantic concepts extracted from the latent representations of LLMs relate to downstream behaviors, such as generating incorrect answers. In doing so, we study two core latent dynamics: *target concept retrieval* and *compositional concept extraction.*

Unfortunately, *existing neural-embedding interpretation methods restrict this joint investigation of input-output features*, due to their constraints on how latent features are defined, typically relying on either the model's output vocabulary or unsupervised extraction. This work introduces *Hyperdimensional Probe*, a novel probing paradigm that combines ideas from symbolic representations and neural probing. Inspired by the sparsity constraints of Sparse Autoencoder (SAEs), we exploit properties of Vector Symbolic Architectures (VSAs) and hypervector algebra to overcome major constraints in prior analysis. Functioning as a *hybrid supervised probe*, our method combines the bounded, interpretable feature space of standard supervised probes with the queryable proxy space of SAE-based methods. It also removes the single-token representation constraint of conventional output-oriented inspection, enabling feature sets with unrestricted abstraction and symbol sources. Thus, our probe combines the supervised learning paradigm of traditional probes with the sparse, dictionary-based representation principle of SAEs. In its core implementation, our approach relies on a unidirectional transformation from neural embeddings to VSA-based representations, which inherently precludes causal validation. While intervention-based analysis falls outside our primary scope, we have conducted experiments with a bidirectional design to enable VSA-based steering (Appendix I.4), showcasing the applicability of VSAs for steering model's text generation.

In particular, our work studies concept-driven dynamics across two distinct experimental settings: completing analogy-style inputs (Section 5), and text generation for Question-Answering (QA; Section 6). Using input-completion tasks, we address the following research questions: whether language models represent all

in-context concepts, key–value associations, or only the output concept (RQ1–RQ3), and whether this representation evidence is consistent across different input types (RQ4) and language models (RQ5). In contrast, the QA-based setup focuses on the dynamics of compositional concept extraction throughout the generation process of language models. This setup investigates the semantic concepts encoded in model's internal state before and after its text generation process, revealing how question- and answer-related concepts evolve and how these observational evidence are associated with its answers (RQ6).

Figure 1 shows our probing framework, from processing neural embedding ($F$) and training encoders ($T$) to output-oriented inspection ($I$). Our work presents both methodological and experimental contributions:

- *Vector Symbolic Architectures for neural probing*: latent features of LLM vector space can be expressed through VSA-based representations and hypervector algebra across diverse language models;

- *Hyperdimensional probe*: a probing method that performs input-aligned feature extraction for output-oriented interpretability (Section 4). It avoids the vocabulary constraints of logit methods (Section 5.3) and provides cleaner results than SAE-based analyses (Section 5.4);

- *Effective compression of LLM embeddings*: it enables probing the full residual stream (Section 4.3), removing the need for the layer-selection stage common in earlier probing work;

- *Observational insights into the semantic concepts encoded during LLM inference*: Section 5 uses template-based tasks to reveal differences in concept-driven compositionality among diverse input types and language models. Section 6 tracks concept-related dynamics immediately before and after the model's text generation in a QA-based inference of two diverse language models.

## 2 Related work

The latent representations of transformers[1] are high-dimensional linear vector spaces that aggregates the outputs of all hidden layers (Elhage et al., 2021). In recent years, three main approaches have been used to study the features encoded in this vector space (Ferrando et al., 2024): (1) supervised probes, (2) SAEs for input-focused feature extraction, and (3) logit-based methods for output distribution inspection.

**Supervised Probing** is a generic mapping paradigm that maps neural embeddings to task-relevant input features, measuring how much information about them is embedded (Tenney et al., 2019). Conventional probes are tailored and trained directly on task-specific features, from syntactical information (Hernández López et al., 2023; Diego Simon et al., 2024), to space-time coordinates (Gurnee & Tegmark, 2023) and truthfulness (Marks & Tegmark, 2023). However, *tailoring probes* to specific tasks limits generalizability, whereas *directly learning targeted features* complicates distinguishing actual information decoding from probe-induced learning (Hewitt & Liang, 2019).

**Sparse AutoEncoders (SAEs)** provide input-oriented feature extraction using a proxy space learned via sparse dictionary learning (Olshausen & Field, 1997), uncovering superposed latent features (Cunningham et al., 2023). An autoencoder reconstructs the residual stream in an unsupervised fashion, enforcing sparsity in its learned representations. Once trained, these serve as a proxy layer for analysis. SAE activated neurons are interpreted via two strategies: identify representative tokens via logit-based methods (Kissane et al., 2024; Dunefsky et al., 2024); and cluster inputs by shared SAE neurons, followed by manual (Jing et al., 2025) or automatic (Bricken et al., 2023; Lieberum et al., 2024) feature naming. The shortcomings of SAEs that hinder output-oriented investigation are threefold: (1) their *unbounded feature space* yields a set of latent features that are difficult to control and align; (2) their *feature-naming* process restricts semantic grounding to either output-vocabulary labels or ambiguous, data-dependent descriptors; and (3) their dependence on *layer-localized representations* requires identifying a task-specific optimal hidden layer.

---

[1]Also known as *residual stream*.

**Logit Attribution** focuses on output distribution inspection by projecting model's embedding space onto its output vocabulary through either (1) the model's unembedding matrix, also known as Direct Logit Attribution (DLA; Logit Lens by Nostalgebraist, 2020); (2) learned affine transformation (Tuned Lens by Belrose et al., 2023); or (3) patched LLM inference (Patchscope by Ghandeharioun et al., 2024). This logit-based paradigm offers insights into the output distribution (Jastrzebski et al., 2017) by generating projected logits at a chosen point in the forward pass, revealing next-token predictions under the assumption that all subsequent layers are bypassed. However, logit-attribution approaches hinder input-aligned feature extraction by (1) relying solely on token-level surface features, which constrains *probing to output-aligned, ELK-style analysis*, and (2) restricting features to single-token representations, *limiting semantic-oriented analysis of higher-level abstractions*.

Functioning as a hybrid supervised probe, *Hyperdimensional Probe* exploits VSAs and hypervector algebra to unify the diverse perspectives of prior methods. Our paradigm integrates (1) the *top-down interpretability* of conventional probes, (2) the SAE's ability to learn a *sparsity-driven proxy space*, and (3) a higher-level, *jointly input–output analytic perspective* that goes beyond conventional logit-based methods. By leveraging VSA-based feature space (Section 3), we avoid the unbounded feature spaces of SAEs and the need for post-hoc feature naming/filtering. Further, our embedding-ingestion algorithm (Section 4.3) *removes the prior methods' requirement of identifying an optimal hidden layer*. Our approach also mitigates the dichotomy between information decoding and probe-induced learning by learning a transformation from LLM embeddings to a controlled proxy space (Section 4.4) rather than predicting latent features directly. Lastly, it addresses the core limitations of conventional output distribution investigations by supporting feature sets of arbitrary abstraction, cardinality, and symbolic origin. Thus, our methodology combines the strengths of prior approaches for joint input-output feature extraction with deeper output-oriented analysis by:

1. *defining a compositional bounded feature space* (probes) rather than unbounded (SAEs);

2. *learning a sparse proxy space* (SAEs) rather than targeting task-specific features (probes);

3. *querying a proxy space* holistically (SAEs), in contrast to standard classification-based probes;

4. inspecting the LLM vector space *without introducing layer selection* (probes, logit, SAEs);

5. *targeting concept-oriented latent features* (VSAs) rather than token-aligned features.

In summary, our approach bridges two traditionally separate paradigms: supervised probing and sparse representation learning. Our contribution lies in integrating VSA-based symbolic representations into supervised probing, rather than offering another variant of conventional probes or SAEs.

## 3 Background

Vector Symbolic Architectures (VSAs)[2] are a computational framework (Schlegel et al., 2022) inspired by cognitive science, increasingly used to map neural representations to human-readable symbols (Hawkins, 2021; Piantadosi et al., 2024). It has been used for diverse tasks ranging from visual problems such as multi-attribute digit recognition (Frady et al., 2020) and Raven's progressive matrices (Hersche et al., 2023) to mechanistic interpretability of language models (Knittel et al., 2024). This symbolic framework is grounded in the core assumption that individual entities or data structures can be represented as random points in a high-dimensional vector space. Leveraging the concentration of measure phenomenon (Kanerva, 2009; Ledoux, 2001), this framework enables the representation of an exponentially large number of distinct concepts using random vectors (a codebook of concepts), nearly orthogonal to each other with high probability. A codebook $\Phi$ maps predefined concepts to hypervectors, which can then be composed into complex representations using orthogonality and hypervector operations.

---

[2]Also known as *Hyperdimensional Computing.*

**VSA codebook.** We adopt the Multiply-Add-Permute architecture (MAP-Bipolar, MAP-B) from VSAs (Schlegel et al., 2022; Gayler, 1998), using bipolar hypervectors in $-1, 1^D$. Dimensionality $D$, typically $10^2$–$10^4$, depends on the number of concepts (Kanerva, 1988) and representation complexity. MAP-B can theoretically encode $2^D$ orthogonal, independent elements (Schlegel et al., 2022). Its codebook $\Phi \in -1, 1^{n_c \times D}$ stores $n_c$ atomic concepts as bipolar random vectors, generated deterministically from seeds to ensure orthogonality and independence. Each vector is associated with a concept, and $\Phi$ enables the evaluation of representations by comparing them with known vectors. Since MAP-B operates in the bipolar domain, cosine similarity is used (Schlegel et al., 2022).

**Hypervector algebra.** The hypervector algebra (Kanerva, 2009) relies on two operations: *binding* and *bundling*, which support representing complex cognitive structures, such as textual propositions, in a *distributed, noise-tolerant* manner (Gayler, 1998; Kanerva, 2009). *Binding* operation ($\odot$) encodes input features with their associated values. For example, it can associate concepts with contextual information, such as (USA $\odot$ dollar). The *bundling* operation ($+$), or superposition, creates set of (contextualized) concepts by combining multiple concepts into one, such as (USA $+$ Mexico). The resulting bundled vector is by design similar to each of its constituents, enabling retrieval. Binding is obtained via Hadamard product (element-wise) while bundling is element-wise sum. Polarization (sign) is typically required after bundling (Kleyko et al., 2020) to maintain the bipolar domain. This process irreversibly blends the parts, diminishing their similarity to the originals in proportion to their number.

Conversely, *unbinding* ($\oslash$) in VSAs recovers elemental vectors from a binding operation by factoring out one vector via multiplication with its inverse (itself in MAP-B).

# 4 Hyperdimensional probe 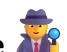

This section introduces our VSA-based framework for extracting information from the neural embeddings of LLMs on analogy-style completion tasks (RQ1–RQ5). Section 4.1 presents an analogy-style corpus as a controlled testbed for testing our method on inputs requiring varied reasoning. Section 4.2 explains how training examples are built using hypervector algebra. We next outline our three-stage pipeline: (1) processing embeddings (Section 4.3, $F$ in Figure 1); (2) mapping them into a controlled proxy space via a neural VSA encoder, producing VSA encodings (Section 4.4, $T$); and (3) extracting concepts from encodings (Section 4.5, $I$). Appendix I outlines the broader applicability of our methodology, including generalization of input representations (I.1), extension to other tasks (I.2), handling out-of-domain data (I.3), and steering model's text generation using a VSA-based paradigm (I.4).

## 4.1 Synthetic corpus

We build a textual dataset to evaluate the core components of our approach within a controlled, and interpretable testbed. Using analogy-style tasks guides LLMs toward concepts and their relations, testing diverse reasoning from syntactic patterns and key-value associations to abstract inference.

**Knowledge bases.** This work focuses on analogies, textual inputs representing pairs of concepts connected by the same type of factual, syntactic, or semantic relationship. We collect pairs of analogies from two knowledge bases: Google analogy test set (Mikolov, 2013), and the Bigger Analogy Test Set (BATS, (Gladkova et al., 2016)). These span 44 domains across five distinct categories, covering a wide range of factual and linguistic relationships, including analogies related to factual knowledge (e.g., a country's currency), semantic relations (e.g., grammatical gender), and morphological modifiers (e.g., verb+men). We also design mathematical analogies using three-digit integers and basic operations such as doubling, cubing, division, and extraction of roots.

**Textual analogies.** After collecting these pairs, we generate 114,099 distinct textual examples, denoted as $\mathcal{S}$, by combining all possible domain pairings. Each training example is formatted as:

$$\texttt{a}_1 \; : \; \texttt{a}_2 \; = \; \texttt{b}_1 \; : \; \texttt{b}_2 \tag{1}$$

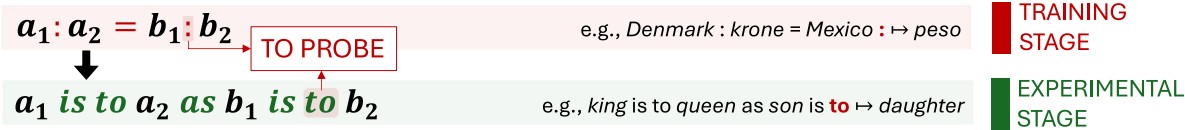

Figure 2: Our experimental setup uses textual inputs with syntactic structures unseen during training.

where $a_1$ and $b_1$ represent the keys of the two pairs, and $a_2$ and $b_2$ are their corresponding values. For example, `Denmark:krone = Mexico:peso` for the countries currencies, and `queen:king = mother:father` for the grammatical gender. Table 14 and Table 15 in Appendix M show the domains grouped by knowledge base and category, respectively. Some concepts span multiple domains, such as `Australia` links to `Canberra`, `English`, and `Australian`. These overlaps can help mitigate the confounding effect of memorizing key-value pairs. For our experiments in Section 5, we further limit confounding effects by using the same pairs but generating a set of textual inputs ($\bar{S}$) with a verbose template: `a₁ is to a₂ as b₁ is to b₂`. Conversely, for training (Section 4.4), we apply data augmentation strategies on $S$, such as key-value swapping, effectively tripling the corpus size which results in 395,944 training inputs (Appendix M).

## 4.2 Training examples

This section describes the process of building VSA-based representations for training input. This procedure, illustrated with our textual templates (Equation 1), generalizes to other templates (e.g., question-answer) or tasks (e.g., toxicity detection; Appendix I) since VSAs and hypervector algebra can encode complex structures across diverse inputs.

**Codebook construction.** The codebook defines the set of all input features; in our case, the contextually relevant concept, and is later used to construct and query VSA encodings. In our controlled setting, the codebook $\Phi$ (feature set) is constructed directly using all unique words included in the corpus, such as: `mexico` $\rightarrow \phi_{mexico} \in \Phi$, and `krone` $\rightarrow \phi_{krone} \in \Phi$. Thus, we create a matrix $\Phi \in \{-1, 1\}^{n_c \times D}$, using the `torch-hd` library (Heddes et al., 2023), where $D$ is the VSA dimension and $n_c = 2,996$ is the number of concepts/features. We set $D = 4096$ as an adequate hidden dimension, given the cardinality of our codebook ($\approx 10^3$), which remains well below the theoretical capacity limit of the MAP-B architecture (Section 3). The average pairwise cosine similarity of the concepts in the codebook is $0 \pm 0.02$, confirming orthogonality (full distribution in Appendix H).

**VSA encodings.** With well-structured textual inputs, extracting input features and building their VSA-based representation is straightforward. Scalability to other input types is addressed in Appendix I.1. For each training input $s \in S$, we generate its encoding by exploiting its constructive words (Equation 1), retrieving their corresponding hypervectors: $\{\phi_{a_1}, \phi_{a_2}, \phi_{b_1}, \phi_{b_2}\} \subset \Phi$. To encode an input sentence, we then exploit hypervector operations: binding and bundling (Section 3). Given that the input template represents two conceptual key–value pairs, we first bind each key to its corresponding value, such as linking each country to its currency in Equation 2. The full text is then encoded through bundling, producing a superposed set of contextualized concepts represented as key–value associations. Ultimately, we polarize it, with the sign function, to maintain the bipolar domain. The input encoding in VSA for a sentence is then computed as:

$$y_s = (\phi_{\text{key}} \odot \phi_{\text{value}}) + (\phi_{\text{key}} \odot \phi_{\text{value}}) + \ldots = (\phi_{a_1} \odot \phi_{a_2}) + (\phi_{b_1} \odot \phi_{b_2}) \tag{2}$$
$$\text{"}\sqcup\text{Denmark}\sqcup:\sqcup\text{krone}\sqcup=\sqcup\text{Mexico}\sqcup:\sqcup\text{peso"} \mapsto (\phi_{\text{denmark}} \odot \phi_{\text{krone}}) + (\phi_{\text{mexico}} \odot \phi_{\text{peso}})$$

## 4.3 Processing neural embeddings $F$

The first stage of our pipeline involves feeding textual inputs to an autoregressive transformer model, followed by obtaining and preprocessing its residual stream ($F$ in Figure 1). Using our corpus, we prompt an LLM with an input sentence $s \in S$, $LLM(s)$. For each textual input, its final word ($b_2$) is removed beforehand as it represents the value of the second analogy, our target concept.

**Caching token embeddings.** Our probing goal is to *inspect the complete internal state of a language model prior to its textual output*, capturing all encoded concepts without assuming their relation to the output. To this end, we examine the residual stream in the final token representation, focusing on the middle to last layers. Emerging evidence shows that transformers encode next-token information in the final token due to their autoregressive nature (Elhage et al., 2021; Olsson et al., 2022), refining it in later residual stream layers (Belrose et al., 2023; Hernandez et al., 2023). Specifically, for an autoregressive language model with $L$ hidden layers, we consider the embeddings (with size $d$) of the last token ("$:$") in the latter half, for all $l \in [L/2, \ldots, L]$, yielding a matrix in $\mathbb{R}^{L/2 \times d}$.

However, considering such a wide range of layers presents a computational challenge, as probing a high-dimensional matrix can significantly increase the computational footprint of the probing pipeline. Further, adjacent layer-wise embeddings are highly correlated (0.9) as shown in Appendix O.1, likely encoding redundant numerical patterns, and thus similar information. Here, we define representation redundancy as the approximate linear dependence among LLM hidden layer embeddings. Appendix O.2 shows that the LLM embedding space is roughly low-rank, with only a few rows/layers (or their combinations) contribute meaningful structure.

**Dimensionality reduction.** To reduce the computational cost of our approach, we lower the encoder's input dimensionality by introducing two dimensionality-reduction steps: *k-means clustering* (Jain & Dubes, 1988), and *sum pooling*. Clustering reduces redundancy in representation by grouping similar vector regions in the embedding space and computing centroids, achieving knowledge distillation. To determine the optimal range for $k$, we adopt the silhouette score (Rousseeuw, 1987). A trade-off between reduction, granularity, and model variability emerges with 3–7 clusters (Appendix O.3). We set $k = 5$ to maintain the essential data structure while supporting effective dimensionality reduction.[3] We then apply sum pooling, which consists of summing all centroid embeddings;[4] merging group representatives (k-dimensional matrix) into a vector exploiting the additivity property of LLM embeddings demonstrated in previous work (Bronzini et al., 2024). For example, these reduction steps allow us to downsize the probed embedding space of OLMo-2:

$$\mathbb{R}^{33 \times 5120} \rightarrow \mathbb{R}^{5120}.$$

Appendix O.5 presents an ablation showing that skipping these two compression steps increases the encoder's trainable parameters tenfold. In summary, the neural representation of a textual input from a language model is processed through the ingestion procedure $F$, as summarized in Algorithm 1.

### 4.4 Neural VSA encoder $T$

We train a supervised model to map token embeddings from an autoregressive transformer into VSA encodings with a known representation ($T$ in Figure 1). We define a supervised regression model $\mathcal{M}$, a shallow feedforward neural network, to map the LLM vector space to bipolar hypervectors. The model $\mathcal{M}$ is trained on the LLM-VSA dataset generated using the corpus $\mathcal{S}$ (Section 4.1), which consists of paired LLM embeddings ($e_s$ in Algorithm 1) and their corresponding VSA representations ($y_s$ in Equation 2). The model infers latent features from the unknown LLM vector space to translate the encoded semantics into VSA representations with explicit and interpretable semantics. We define the *neural VSA encoder* model $\mathcal{M}$ as a three-layer MLP with 55M–71M parameters (depending on the input embedding size $d$; see Appendix C), performing a non-linear transformation:

$$\mathcal{M} : \mathbb{R}^d \rightarrow \{-1, 1\}^D, \quad e_s \rightarrow y_s. \tag{3}$$

We use the hyperbolic tangent function (`tanh`) in the output layer for bipolar outputs and incorporate residual connections to enhance training stability and convergence. The training process minimizes the Binary Cross-Entropy (BCE) error between the bipolar target hypervectors and the predictions. To ensure compatibility with such binary loss function, targets are temporarily converted to binary based on their sign; and predictions are smoothly mapped to the range [0, 1] using the sigmoid function. A Mean Squared Error

---

[3]Appendix O.4 shows that the clusters consistently group adjacent layers.
[4]Preliminary evidence suggests that directly summing all layers (up to 33) results in a noisier representation.

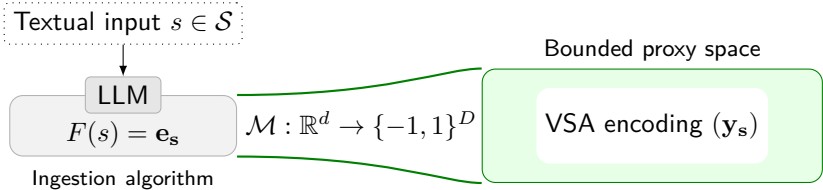

Figure 3: The regression model that maps the neural representations into a controlled vector space.

(MSE) regularization term is added to the loss, with a coefficient of 0.1.[5] Implementation details for the training process are reported in Appendix D.1.

**Language models.** We validate our methodology on embeddings from popular open-weight LLMs available on Hugging Face with 355M-109B parameters, experimenting with different embedding sizes and layer counts. In particular, we test the latest Meta AI's Llama 4 Scout, (Meta AI, 2025) a multi-modal mixture of 16 experts (MoE), Llama 3.1-8B (Grattafiori et al., 2024), Microsoft's Phi-4 (Abdin et al., 2024), EleutherAI's Pythia-1.4b (Biderman et al., 2023), AllenAI's OLMo-2-32B (OLMo et al., 2024), and OpenAI's legacy GPT-2-medium (Radford et al., 2019).

**Performance.** The LLM-VSA dataset uses a random 70-15-15 split of $\mathcal{S}$ for training, validation, and test sets. Since our setting can be interpreted both as a vector-based regression task and a multi-label classification problem,[6] we evaluate our approach using two distinct metrics: cosine similarity and multi-label binary accuracy. For binary accuracy, targets and predictions are binarized based on sign. First, evaluating the cosine similarity between the predicted and target VSA encodings yields a test-set average score of 0.89 (best LLM in Appendix D, Llama 3.1-8B), indicating strong numerical alignment between our encoder's outputs and the target encodings. Second, we obtain an average binary accuracy of 0.94, which indicates robust classification accuracy after polarizing the predictions with the sign function. This means that on average, the VSA encodings produced by our trained model deviated from the targets by only 6% of the vector elements, a negligible error given VSA's large tolerance to noise. All tested models exhibit consistent performance;[7] layer count has no effect, whereas reducing the embedding dimension is found to be slightly detrimental. Appendix D also includes a baseline that compares performance against a null model using randomly permuted input embeddings. Because its test-set predictions collapse toward a nearly constant value near the lower bound of the bipolar domain ($AVG = -0.88 \pm 0.04$), indicating a low-information and weakly discriminative representation, we observe cosine similarities around 0.5. We therefore treat values near 0.5 as the empirical null-signal baseline for this setup. These results support our empirical hypothesis that the latent features encoded in the neural embedding of LLMs (RQ7) can be faithfully captured using VSA-based representations (MAP-B) and hypervector algebra.

### 4.5 Probing VSA encodings $I$

In the third stage of our pipeline ($I$ in Figure 1, Section 5), we examine the VSA encodings produced by our trained neural VSA encoder $\mathcal{M}$, extracting the embedded concepts. To retrieve the embedded atomic concepts, we use the *unbinding* operation from VSA algebra ($\oslash$, Section 3). This vector operation reverses binding, which in our case links a pair's key with its corresponding value, enabling one vector to be extracted from another. Since the generated VSA encoding may encode either no or several concepts, we attempt to extract the target concept ($b_2$) by dynamically testing the unbinding operation with various candidates.

*This concept-related flexibility represents the novelty and added value of VSA-based probing*, allowing us to query our proxy space without prior assumptions on the number of concepts. Consequently, we distinguish between two scenarios: in the first, no unbinding operations are required when the model encodes none or a single concept; in the second scenario, when multiple concepts are embedded, we test the unbinding

---

[5]Empirical results demonstrated better performance than other coefficients tested, ranging from 0.01 to 1.

[6]VSA encodings can be viewed as vectors with $D$ distinct labels, each assuming one of two possible values.

[7]Appendix D reports the training performance of our neural VSA encoder $\mathcal{M}$ for all of the six models.

operation with different concepts to isolate a single one. For example, unbinding a VSA encoding with the concept of `Mexico` and obtaining `Peso` suggests that the probed encoding originally incorporated both the key and value of the target analogy pair:

$$
\begin{aligned}
\text{LET} \quad & s := \text{``}\texttt{Denmark:krone=Mexico:}\text{''} \mapsto \text{``peso''} \\
\text{COMPUTE} \quad & y_s = \mathcal{M}(F(s)) \\
\text{QUERY} \quad & y_s \oslash \phi_{\text{mexico}} = \phi_{\text{peso}} + \text{noise} \\
\text{THEN} \quad & y_s \approx (\phi_{\text{mexico}} \odot \phi_{\text{peso}})
\end{aligned}
\tag{4}
$$

When probing an encoding ($y_s$ in Equation 4), we pick in-context concepts ($\phi_{\text{denmark}}$, $\phi_{\text{krone}}$, and $\phi_{\text{mexico}}$), and their combinations, as candidates for unbinding. The best candidate was chosen by benchmarking the resulting concept after unbinding, against the in-context and target concepts through cosine similarity with a threshold for concept detection equal to 0.1. Empirical tests show relevant concepts exceeded this low threshold, while noisy ones remain below, likely due to VSA's noise tolerance. If no concepts were detected, unbinding was skipped. In the experiments reported in Section 5, 80% of unbinding operations, averaged across all models, relied on the key of the target pair. In contrast, no operation was applied in 12% of the cases. Appendix E shows the proportions of other candidates and highlights the variations among models. Meanwhile, the *unrelated baseline* in Appendix F shows results of unbinding with concept candidates that do not relate to the input.

## 5 Experiments on input-completion tasks

This section uses controlled, template-based analogy tasks (Section 4.1) to investigate the differences in concept-driven compositionality (RQ1–RQ3) among diverse input types (RQ4) and language models (RQ5), uncovering the semantic concepts encoded in the model's internal state when tasked with input completion. For this input-completion setup, our work studies two concept-oriented aspects using our VSA-based probe: *target concept retrieval*, and *compositional concept extraction*. They address two complementary questions: whether the model's internal state represents the target output concept (Figure 4) and what semantic concepts compose that internal representation (Table 1).

We begin by outlining the analogy-oriented experimental setup in Section 5.1. We then present our findings on concept extraction using the proposed probe in Section 5.2. Next, Section 5.3 compares these results with a traditional logit-based investigation (i.e., Direct Logit Attribution), which exhibits more superficial probing capabilities, likely because it depends on the model's vocabulary space and, thus, token-level latent features. Lastly, Section 5.4 reports an experimental comparison with Sparse Autoencoders (SAEs).

### 5.1 Experimental setup

**Data.** This work adopts several strategies to mitigate the confounding effects of learning regularities from the training stage (Section 4.4). The first is to test the trained neural VSA encoders, $\mathcal{M}$, on textual inputs formatted with syntactic structures ($\bar{\mathcal{S}}$) that differ from those seen during the training stage ($\mathcal{S}$, with $\bar{\mathcal{S}} \neq \mathcal{S}$). Specifically, experiments are conducted using the verbose template described in Section 4.1. Consequently, we perform information decoding from the vector space of different token representations, shifting from the colon token of the training examples (Equation 1) to the token `to` in $\bar{\mathcal{S}}$. This strategy, which can be seen as a held-out test with respect to the distribution of LLM embeddings learned by the encoder, further mitigates confounding effects caused by probe-induced learning, which are more pronounced in conventional supervised probes.

**Metrics.** For target concept retrieval, our experimental evaluation has a two-fold objective (Equation 4): we assess the performance of LLMs in next-token prediction and our VSA-based probing method for retrieving targets from their latent representations using *precision@k*. We measure the LLM's performance via binary precision on next token prediction against the target word; softmax score of the most likely next token and the target one; and the rank of the target token on the ordered softmax scores. The downstream performance of LLMs is measured using the most likely next token (*next token@1*) and the top-5 tokens (*next token@5*),

Table 1: **Compositional concept extraction for the input-completion tasks.** Each entry lists all the concepts extracted by the hyperdimensional probe. `Key | Target` indicates extraction of the key ($b_1$) and value ($b_2$) of the target analogy; `Key` for only $b_1$. `Example` refers to $a_1$ and $a_2$, the in-context example's concepts; `Context | Target` for all concepts. `Out-of-context` indicates concepts unrelated to input, `Key Values` refer to analogy keys originating from a different domain. `NONE` means no concepts. *Our probe captures target-aligned combinations in roughly 80% of cases.*

| Extracted Concepts (%) | GPT-2 | Pythia | Llama 3.1 | Phi-4 | OLMo-2 | Llama4, Scout | AVERAGE |
|---|---|---|---|---|---|---|---|
| Key \| **Target** | 60.0 | 66.9 | **85.4** | 84.8 | 80.1 | 79.0 | 76.0 ± 9.4 |
| NONE | **21.9** | 16.7 | 6.9 | 7.6 | 8.5 | 11.5 | 12.2 ± 5.4 |
| Key | 4.5 | **6.1** | 0.6 | 1.0 | 1.8 | 3.4 | 2.9 ± 2.0 |
| Example | **5.8** | 2.4 | 1.5 | 1.3 | 2.0 | 0.8 | 2.3 ± 1.6 |
| Context \| **Target** | 1.1 | 1.9 | 1.5 | 1.2 | **4.4** | 2.4 | 2.1 ± 1.1 |
| Key \| Key Values | 1.3 | 1.4 | **1.8** | 1.5 | 1.1 | 0.8 | 1.3 ± 0.3 |
| Out-of-context | **1.6** | 1.2 | 0.5 | 0.7 | 0.8 | 1.1 | 1.0 ± 0.4 |
| Example Value \| Key Values | **1.5** | 1.3 | 0.3 | 0.0 | 0.2 | 0.1 | 0.6 ± 0.6 |
| Key Values \| **Target** | 0.4 | 0.1 | 0.3 | **0.4** | 0.1 | 0.1 | 0.2 ± 0.1 |
| **Target** | 0.1 | 0.1 | 0.1 | **0.2** | 0.1 | 0.1 | 0.1 ± 0.0 |

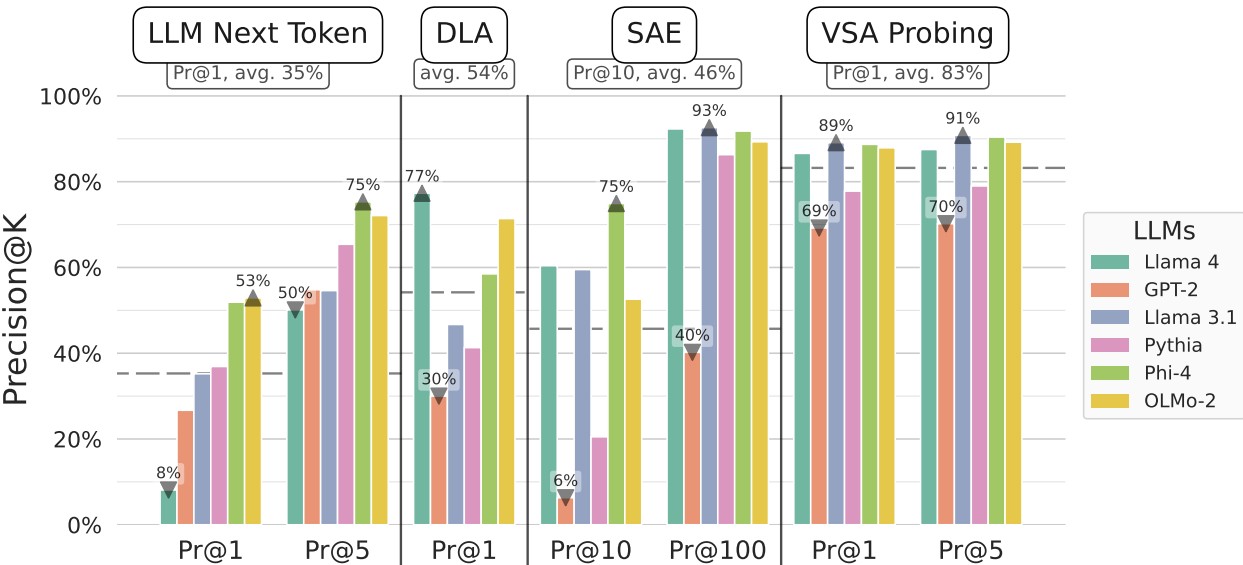

Figure 4: **Target concept retrieval for the input-completion tasks.** LLM performance on analogy completion with targets (left) and our decoding method's ability to extract targets from LLM embeddings (right), with logit-based and SAE-based results shown in the middle.

capturing uncertainty in the model's softmax distribution. To evaluate the performance of our VSA-based probing approach, we assess the binary precision of retrieving the target VSA concept from LLM latent representations via *probing@1*, and *probing@5*.

## 5.2 Concept extraction

Figure 4 reports target concept retrieval using VSA-based probing, LLM next-token prediction performance, and comparisons with logit-based and SAE-based analyses. Appendix F tabulates the same results with variability and two control experiments. Specifically, we conduct two validation experiments (F.1) to verify that the proposed approach decodes information from LLM embeddings rather than producing meaningful outputs under arbitrary conditions: a null model using randomly permuted input embeddings, and probing the learned space with concepts unrelated to the input.

On the other hand, Table 1 presents findings for the compositional concept extraction investigation, with each row showing the unique concept sets produced by the extraction procedure of Section 4.5, the com-

prehensive exploration of the VSA-based proxy space. Each entry lists all concepts involved in the probing procedure: the concepts used for the unbinding step and the concepts recovered afterwards. For the example in Equation 4, the mappings are defined as: $\phi_{\text{mexico}} \mapsto$ Key, $\phi_{\text{peso}} \mapsto$ Target, $\{\phi_{\text{denmark}}, \phi_{\text{krone}}\} \mapsto$ Example, and $\{\phi_{\text{denmark}}, \phi_{\text{krone}}, \phi_{\text{mexico}}\} \mapsto$ Context.

**VSA probing exposes diverse concept-oriented compositionality (RQ1–RQ5).** Regarding the concepts extracted by our trained probe, we achieve an average probing@1 across all models equal to 83% (right side of Figure 4, Appendix F), recovering the target concept together with its corresponding key in most cases (60% for GPT-2 and 85% for Llama 3.1; Key | Target in Table 1). Notably, GPT-2 shows the highest proportion of cases with no concepts extracted (22%; NONE) followed by cases extracting only the concepts of the input example (6%; Example), and ranks second in extracting only the target concept keys (5%; Key). Thus, its internal states largely fall back on input-related concepts, *reflecting limited understanding needed to complete the analogy-style input correctly.* On the other hand, OLMo-2 has the highest proportion of instances in which our probing approach retrieves the target concept alongside all in-context concepts (4%, Context | Target), *indicating its richer representation in its final state for both the input context and next word.* This latent richness is then reflected in its performance on next-token prediction, achieving the highest next token@1 equal to 48% (Figure 4). In cases where the target word was not among the top five predictions of Llama 4 (Appendix L.1; 28% for OLMo-2 in Appendix L.2), our probing method successfully extracted the target concept and its associated key in 70% of instances, while no concept was retrieved in 18% of cases (26% for OLMo-2). Although the first findings supports previous experimental evidence, the absence of extracted concepts merits a more granular analysis across the varied input-completion tasks, the domains of our analogy-style inputs (RQ4; see also Table 15 in Appendix M).

**Variable compositional concept extraction in input-completion tasks.** We examine probe performance across different types of textual inputs, defining success and failure by the presence or absence of extracted concepts. Table 2 presents the distribution of instances where no concepts were extracted, categorized by input type. While we observe some variation across language models, these empirical findings reveal consistent evidence of empty representations, which may be tied to the specific type of reasoning required:

1. **Linguistic analogies** yield the lowest rate of missing concept extraction (1–1.8%), suggesting richer LLM representations, likely due to reliance on all concepts to capture *implicit syntactic patterns.*

2. **Factual knowledge** and **semantic relations** show slightly higher but still low blank rates (5.3–7%). Since these analogies rely on *key–value associations*, blanks may reflect missing associations in the model.

3. **Semantic hierarchies** (34.8%) and **mathematical analogies** (89.5%) yield the highest blank rates. Both require more *abstract reasoning*, but the large gap in mathematics likely stems from the rarity of analogical tasks with numbers, compared to equation solving or standard math problems more common in training data.

Table 2: **Proportion of instances with no retrieved concepts across diverse input types for two language models.** Lower values imply richer semantic embeddings. OLMo2 yields fewer conceptually blank representations in most categories, though Llama 4 holds a slight advantage in mathematics and grammar.

| Area | Llama 4 (docs, %) | OLMo2 (docs, %) | AVG | Sample | Domain |
|---|---|---|---|---|---|
| Mathematics | **87.8** | 91.1 | 89.5 | 80 is to 160 as 98 is to | math double |
| Semantic Hierarchies | 38.8 | **30.8** | 34.8 | limousine is to car as monorail is to | hyponyms |
| Semantic Relations | 10.0 | **3.9** | 7.0 | Croatia is to Croatian as Switzerland is to | nationality adjective |
| Factual Knowledge | 5.5 | **5.1** | 5.3 | euclid is to Greek as galilei is to | name nationality |
| Verbal & Grammatical Forms | **1.4** | 2.1 | 1.9 | seeing is to saw as describing is to | past tense |
| Morphological Modifiers | 1.1 | **0.8** | 1.0 | agree is to agreement as excite is to | verb+ment |

Table 3: **Concepts extracted though VSA-based probing when DLA yields no concepts.** The table highlights VSA can also capture model's variability (e.g., in-context concepts, target concepts). The table highlights key shared items across models, covering nearly 98% of all extracted concepts.

| | GPT-2 | Pythia | Llama 3.1 | Phi-4 | OLMo-2 | Llama 4, Scout | AVERAGE |
|---|---|---|---|---|---|---|---|
| NONE: no concepts from DLA (%; $\mathcal{D} \subset \bar{\mathcal{S}}$) | 33.9 | 32.8 | **47.4** | 33.1 | 14.6 | 15.4 | $29.5 \pm 11.4$ |
| Concepts extracted from $\mathcal{D}$ by VSA (%) | | | | | | | |
| Key \| **Target** | 53.5 | 56.5 | **76.6** | 70.5 | 44.5 | 42.6 | $57.4 \pm 12.5$ |
| NONE | 26.8 | 24.3 | 13.7 | 18.6 | 41.0 | **43.2** | $27.9 \pm 10.9$ |
| Example | **6.8** | 3.0 | 2.1 | 2.1 | 3.4 | 2.0 | $3.2 \pm 1.7$ |
| Key | 5.2 | **7.3** | 0.7 | 0.9 | 1.6 | 3.0 | $3.1 \pm 2.4$ |
| Out-of-context | 2.0 | 1.8 | 1.1 | 1.9 | 3.8 | **4.5** | $2.5 \pm 1.2$ |
| Key \| Pair Values | 1.8 | 2.1 | 2.5 | **3.2** | 2.4 | 1.8 | $2.3 \pm 0.5$ |
| Context \| **Target** | 0.7 | 1.2 | 1.1 | 0.7 | **1.9** | 1.4 | $1.2 \pm 0.4$ |
| **Target** | 0.1 | 0.1 | **0.2** | 0.0 | 0.1 | 0.0 | $0.1 \pm 0.1$ |

**High variability in LLMs' next-token predictions.** Concerning LLM next-token predictions performance in input-completion tasks, in an unexpected contrast, the largest model evaluated (109B; Llama 4, Scout) exhibited the lowest precision@1 in the next-token prediction task (8%, Figure 4), even underperforming the legacy GPT-2. Yet, its next-token@5 was comparable to others (still the lowest), but ranked among the best in probing@1. *Strong probing performance suggests the final state encodes the target concept, but the model often fails to output it.* This might be caused by exogenous (e.g., prompt design) and endogenous factors (e.g., tokenization). As shown in Appendix F.2, the model frequently predicted a space instead of the correct word, which still often appeared in its top five predictions.

This highlights the potential variability of conventional token-oriented analyses, such as logit-style probes, arising from specific prompt designs and model-specific tokenization. In contrast, under the same zero-shot prompting setting, our VSA-based analysis remains robust to these exogenous and endogenous factors.

### 5.3 Comparison with Logit Attribution

For validation, we first compare our VSA-based findings with conventional output distribution inspection. Specifically, we apply Direct Logit Attribution (DLA/Logit Lens; Section 2) to the neural embeddings of all models using $\bar{S}$, emphasizing the constraints of logit-based analysis imposed by the model's token vocabulary.

We use fuzzy token-to-concept matching with our concept set (e.g., "pes" $\mapsto$ peso), and consider projected next-token predictions from the model's middle to last layers (Appendix G.2) of the last token, as VSA probing. DLA produces no concepts in nearly 30% of analogies on average ($\mathcal{D}$ in Table 3; +17% compared to VSA in Table 1). It also exhibits lower performance and greater variability in target concept retrieval, with a similarly large drop in comparison to VSA-based precision@1 (Figure 4). In instances without concepts from DLA ($\mathcal{D}$ in Table 3), our VSA probing extracts, on average, the key-target pair in 57% of all analogies, while returning none for 28%. For instance, for the analogy king is to queen as son is to $\mapsto$ daughter, using OLMo-2, our probe extracts the key-target concepts (son and daughter), while DLA produces no concepts. The model predicts the next token as ? with a softmax score of 0.06, followed by father (0.05); the target word has a rank of 37. Since conventional output inspections operate within the model's vocabulary space, *token-level logit analyses often produce shallower findings and show greater variability* caused likely by exogenous factors, such as prompt design, or endogenous ones.

### 5.4 Comparison with Sparse Autoencoders

Lastly, we compare our VSA-based results with those derived from an analysis using Sparse Autoencoders (SAEs; Section 2). This section describes our experimental protocol for SAE-based analysis in our input-completion setting, from training autoencoders and performing feature naming, to conducting a compositional concept extraction investigation. Appendix G further discusses our experimental comparison choices.

**Training.** We train autoencoders on neural embeddings from each of the six language models, using the training corpus $\mathcal{S}$. Input embeddings are ingested using the same procedure as the VSA-based method (Section 4.3; Algorithm 1), which eliminates the need for the conventional search over the optimal model's hidden layer. We define a shallow encoder-decoder neural network to reconstruct LLM's embeddings with a sparsity constraint in its latent representations. Specifically, the SAE's encoder is a fully-connected linear layer with a ReLU activation function, while the decoder has a single linear layer. The SAE's latent dimensionality is defined as a fixed multiple of the input embedding dimension, specifically a factor of 4. For instance, the SAE for GPT-2 uses a latent size of 4096 ($1024 \times 4$), whereas OLMo-2's uses 20480 ($5120 \times 4$). Regarding sparsity, we adopt a top-k sparsity constraint in the forward process (Makhzani & Frey, 2013), where $k$ is set to a fixed proportion of SAE latent dimensionality (2%, e.g., $k = 82$ for GPT-2, and $k = 410$ for OLMo-2). The training process minimizes the Mean Squared Error (MSE) between the LLM embeddings and the reconstructed ones. L1 regularization is also added to the loss function to further encourage sparsity during training. Using a random 70–15–15 split of $\mathcal{S}$ into training, validation, and test sets, the SAEs obtain an average cosine similarity of 0.98 between the input embeddings and their reconstructions on the test set (Appendix G.4). Following the VSA-based experimental setup (Section 5.1), the trained SAEs are subsequently tested on textual inputs with syntax differing from that used during training ($\bar{\mathcal{S}}$, with $\bar{\mathcal{S}} \neq \mathcal{S}$).

**Feature naming.** The feature naming stage uses a logit-based approach. SAE-activated neurons are interpreted via Direct Logit Attribution, projecting the SAE space into the LLM's output vocabulary using the SAE decoder and LLM's unembedding layer (Elhage et al., 2022). Specifically, each activated neuron is associated with the top-10 tokens from its induced logit distribution, which are then compared to our concept set using the fuzzy token-to-concept matching function (e.g., "extrov" $\mapsto$ extrovert) from Section 5.3.

**Target concept retrieval.** We initially evaluate SAE's ability to extract target concepts from LLM embeddings by checking whether the targets appear among the top-k most strongly activated SAE neurons. Unlike VSA- and logit-based probing, SAEs adopt a bottom-up approach that uncovers an unbounded set of latent features without inherent relevance ranking or filtering, limiting alignment with our bounded input–output concept framework. Consequently, we extend the measurement scope of the previous analysis by considering concepts derived from the top-10 and top-100 SAE neurons, reported as precision@10 and precision@100, respectively. In general, the SAEs achieve an average precision@10 across all models equal to 46% (Figure 4 and Table 8), reaching comparable performance to VSA probing (probing@1) only when considering the top-100 activated SAE neurons (precision@100).

**Compositional concept extraction.** Regarding the composition of the concepts extracted from the top-10 most strongly activated SAE neurons (Table 4), the majority of the instances (43%) across all language models produce only concepts that are not referred to the four in-context concepts for inputs (out-of-context in Table 4). This is followed by the extraction of the key with target concepts (with fur-

Table 4: **Compositional concept extraction with SAEs using the top-10 most strongly activated SAE neurons.** Key|Target indicates extraction of the key ($b_1$) and value ($b_2$) of the target analogy; Key for only $b_1$. Example refers to $a_1$ and $a_2$, the in-context example's concepts; Context|Target for all concepts. Out-of-context indicates concepts not referred to the four in-context concepts.

| Extracted Concepts (%) | GPT-2 | Pythia | Llama 3.1 | Phi-4 | OLMo-2 | Llama 4, Scout | AVERAGE |
|---|---|---|---|---|---|---|---|
| Out-Of-Context | **83.9** | 61.5 | 31.3 | 14.6 | 39.2 | 29.3 | 43.3 |
| Key \| Target \| Out-Of-Context | 2.6 | 11.7 | 45.4 | **54.1** | 37.5 | 42.3 | 32.3 |
| Target \| Out-Of-Context | 2.9 | 4.5 | 7.2 | 8.7 | 4.9 | **9.0** | 6.2 |
| Key \| Out-Of-Context | 2.6 | 4.4 | 5.3 | **6.2** | 4.2 | 5.4 | 4.7 |
| Context \| Target \| Out-Of-Context | 0.0 | 1.7 | 0.0 | 3.5 | **4.8** | 3.6 | 2.3 |
| Example \| Out-Of-Context | 2.2 | **7.1** | 1.0 | 0.8 | 1.4 | 1.3 | 2.3 |
| Example Value \| Key \| Target \| Out-Of-Context | 0.0 | 0.5 | 0.0 | **4.5** | 2.3 | 2.5 | 1.6 |
| Example Value \| Out-Of-Context | 2.4 | **3.0** | 1.3 | 1.5 | 1.0 | 1.8 | 1.8 |
| Example Key \| Out-Of-Context | **2.1** | **2.1** | 0.5 | 0.5 | 0.5 | 0.6 | 1.1 |
| Example Value \| Target \| Out-Of-Context | 0.0 | 1.1 | 0.0 | **2.1** | 1.0 | 1.6 | 1.0 |
| Example Key \| Key \| Target \| Out-Of-Context | 0.1 | 0.4 | 1.1 | 1.5 | **1.7** | 0.9 | 0.9 |

ther out-of-context concepts) in about 32% of the instances, whereas key concepts or target concepts (with further out-of-context concepts) is found respectively in 6.2% and 4.7% of the experimental instances. For instance, in the earlier example (*king is to queen as son is to*), the SAE-based analysis on OLMo-2 detects only concepts that lie outside the bounded in-context concept set (*out-of-context*), surfacing concepts such as man, human, girl, boy, and gender. This prevents direct answers to our research questions about in-context input concepts (RQ1-RQ3), but it aligns with the input type (gender analogy) and prior output distribution findings, where *father* is among the model's most likely predictions for this input (see Section 5.3).

While this suggests that SAEs are capable of capturing a broad range of latent features in LLMs, as VSA-based probing, it also highlights an important difference in their operational characteristics. In particular, SAEs lack explicit relevance filtering or task-specific targeting during feature extraction. In contrast to logit- and VSA-based methods, which are constrained by next-token representations and predefined concept sets, respectively; SAEs operate bottom-up, uncovering a wide and potentially unbounded set of features without intrinsic task-aligned ranking. As a result, task-relevant features (e.g., in-context concepts in analogy inputs) may co-occur with many unrelated features, affecting alignment with a subsequent investigation within a task-oriented concept space. Additionally, the choice of $k$ in selecting top-k SAE activations introduces a tuning parameter that can influence evaluations and complicate comparisons with methods such as VSA probing that do not require such selection.

The absence of relevance-based feature filtering, stemming from SAEs' objective of uncovering latent structure in an unsupervised manner, can ultimately lead to *noisier interpretability analyses in bounded conceptual feature spaces.*, such as those required by our investigation. In summary, VSA probing is particularly effective in experimental settings that require a constrained or structured conceptual feature space.

# 6 From input-completion tasks to text generation

After validating our methodology with controlled input-completion tasks in Section 5, this section investigates the evolution of the model's internal state throughout text generation in a Question-Answering (QA) setting. This evaluation has two objectives: to investigate concept-related dynamics before and after generation during QA-based inference across diverse language models (RQ6), and to further assess the generality of VSA-based probing in a substantially different experimental setting targeting a distinct probing objective. Although this setting applies the same pipeline introduced in Section 4, the two settings construct and probe the VSA proxy space through fundamentally different strategies. The neural VSA encoder used in the input-completion setting learns a VSA-based proxy space from a fixed, predefined structure of target latent features. In contrast, the encoder in this setting learns a proxy space with features grounded in lexical semantics, encoding a variable number of concepts through dynamic representations. This complementary evaluation demonstrates that our approach is not tied to a specific proxy-space construction strategy or template-based conceptual representation.

**Training data.** We exploit the Stanford Question Answering Dataset (SQuAD, Rajpurkar et al. 2016) to study concept-oriented dynamics throughout the generation process of two different language models, Llama 3.1-8B and GPT-2-medium. This setting also enables us to compare the semantic concepts extracted through VSA-based probing with those associated with the input questions and their corresponding answers. This dataset evaluates extractive QA, where each answer is a text span within a given context, using questions written by crowdworkers based on Wikipedia articles. This aligns with our concept-focused probing, as questions and answers point to contextual semantic concepts, allowing us to benchmark feature extraction against features derived from both.

Unlike the predefined feature structure of Sections 4 and 5, this setting automatically extracts question- and answer-related concepts for each textual input grounded in their lexical semantics. This feature extraction process identifies nouns, verbs, and adjectives using two well-known lexical knowledge bases, Word-Net (Miller, 1995) and DBpedia (Lehmann et al., 2015). Lexical semantics serves as the set of all possible latent features for this setting, our VSA codebook $\Phi$. Using bundling operations, we then construct 634,368

training examples $\mathcal{Q}$ by progressively pairing SQuAD questions with their associated lexical features:

$$
\begin{aligned}
\text{``What was the } name\text{''} &\mapsto (\phi_{\text{name}}) \\
\text{``What was the } name \text{ of the } ship\text{''} &\mapsto (\phi_{\text{name}} + \phi_{\text{ship}}) \\
\text{``What was the } name \text{ of the } ship \text{ that } Napoleon\text{''} &\mapsto (\phi_{\text{name}} + \phi_{\text{ship}} + \phi_{\text{napoleon}}) \\
\cdots
\end{aligned} \tag{5}
$$

The neural VSA encoder trained on this QA-based setting achieves a test-set cosine similarity of 0.53, and a binary accuracy of 0.74 for mapping embeddings of Llama 3.1 into these VSA encodings. The null model for this setup, with randomly-permuted input embeddings, achieves a cosine similarity of 0.24.

**Probing stage.** For our probing experiments, we consider 10,000 sampled items $\bar{\mathcal{Q}}$ from SQuAD, each formatted as a question preceded by its contextual text (Appendix J; e.g., *"Napoleon III responded with a show of... Q: What was the name of...? A:"*), applying our trained probe on textual inputs that differ from those seen during its training stage as in Section 5. We then apply our trained probe on the model's internal state both before and after text generation, comparing the generated VSA encodings, our proxy space, with the codebook $\Phi$ using cosine similarity with a concept detection threshold of 0.1, as unbinding is not required. Our probe extracts an average of three concepts before both the first and last token generation. In the underlying QA task, the language model achieved an average token-based F1 score of 0.69 (95% CI: 0.68–0.70), and 68% of the generated outputs (95% CI: 67–69) contained the target answer.

As the probing objective shifts from detecting four predefined concepts in LLM latent representations (Section 5) to comparing extracted concepts with those associated with the input questions and their corresponding answers, the evaluation likewise shifts from binary concept detection to semantic concept relevance. Specifically, we evaluate semantic-based concept relevance by computing cosine similarity among the concept embeddings of the concepts extracted from LLM latent representations and question-answer concepts.

**Llama's wrong answers might reflect a loss of focus on the question.** Concerning Llama 3.1, the average similarity of extracted concepts related to the question decreases after text generation: by 4.8% for the entire sample and by 8.0% in the LLM error subset (32% of the sample), while no significant differences are observed prior to generation (Figure 5; left). For answer-related concepts, overall no change is observed before and after text generation, but the LLM error subset shows a slight increase (+3.2–3.5%; Figure 5, right). *This suggests that* these model's failures *may stem from losing focus on the question rather than from a lack of answer-related knowledge.* This hypothesis is further supported by a weak positive Spearman correlation (0.2 with a p-value of $1e^{-99}$; Appendix K) between the model's token-based F1 score and the proportion of question-related concepts extracted after text generation. For example, for the SQuAD query "What do laboratories *try* to *produce hydrogen* from?" (target answer: "*solar energy* and *water*"), the model erroneously outputs "*water* and heat" (F1 = 0.57). Before the model's text generation, our proposed approach extracts the concepts $\phi_{\text{try}}$, $\phi_{\text{produce}}$, $\phi_{\text{hydrogen}}$ (question) and $\phi_{\text{solar}}$, $\phi_{\text{water}}$ (answer); after generation, the question-related concept set reduced to $\phi_{\text{produce}}$ and the answer set gained the concept $\phi_{\text{energy}}$. During text generation, the model lost the concept of hydrogen while refining concepts related to

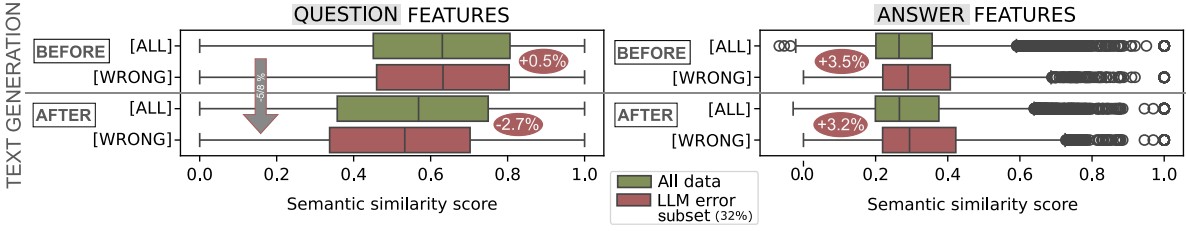

Figure 5: Concepts extracted *before* and *after* the text generation of Llama 3.1, with respect to *question* and *answer* features. Red denotes the subset of failure instances, while green the full sample $\bar{\mathcal{Q}}$.

the answer. This leads to a response that no longer focuses on the question's subject, but reflects the more general concepts still active in the model's internal state.

**GPT-2's state reveals an overrepresentation of input-related concepts.** To contrast the QA-related findings from the encoder trained on a medium-scale language model (Llama 3.1; 8B parameters), we also study the concept-oriented dynamics of a smaller language model with 355M parameters (GPT-2-medium). Regarding its QA-related performance, GPT-2 achieves an average token-based F1 score of 0.133 (95% CI: 0.13–0.14), and 21% of the outputs (95% CI: 20–22) contained the target answers. This poor performance indicates a lack of understanding in the downstream task. In contrast, the encoder trained on its neural embeddings achieves a test-set cosine similarity of 0.52 and a binary accuracy of 0.74, comparable to Llama's training performance. We then apply the trained probe to $\bar{\mathcal{Q}}$, probing the model's internal state both before and after text generation. Our probe extracts an average of six concepts before both the first and last token generation. The average semantic-oriented similarity of extracted concepts decreases after text generation by 13.7% for question-related concepts and 4.5% for answer-related concepts (Appendix J.1). Unlike the previous observational evidence, no concept-based differences are observed in the model's error subset (79%), suggesting erroneous answers of GPT-2 may stem from a different reason than Llama's shifted focus. For the aforementioned example (i.e., "What do laboratories *try* to *produce hydrogen* from?"), GPT-2 erroneously outputs "the most important step in the process is to use a large amount of *water* to produce" (F1 = 0.11). Before the model generates text, our probe identifies the concepts $\phi_{\text{produce}}$ (question), $\phi_{\text{solar}}$ (answer), and the out-of-domain concepts $\phi_{\text{name}}, \phi_{\text{process}}, \phi_{\text{use}}, \phi_{\text{work}}$. After generation, the answer concept shifts to $\phi_{\text{water}}$, no question-related concepts are detected, and the out-of-domain set changes, gaining $\phi_{\text{use}}, \phi_{\text{one}}, \phi_{\text{main}}, \phi_{\text{vacuum}}$ while losing $\phi_{\text{name}}$ and $\phi_{\text{work}}$. While losing question-related concepts aligns with Llama's findings, the wide set of out-of-domain, yet topic-related concepts confirms earlier observations of GPT-2's latent representations when processing analogy-style inputs (Section 5.2). The model's internal representations tend to fall back on concepts closely associated with the input text (e.g., *use*, *process*, *name*), suggesting a shallow grasp of the underlying task; namely, answering with the energy sources used for hydrogen production. This also corresponds with the model's poor performance on this QA-based task.

## 7 Conclusions

This work provides empirical evidence that the latent features of LLM's neural embeddings can be faithfully captured using VSA-based representations (RQ7; Section 4.4). Building on this, VSA probing enables a unified, concept-oriented analysis of features aligned with both inputs and outputs, integrating the complementary perspectives of prior neural probing approaches to uncover non-trivial insights into LLM inference.

The proposed method combines the top-down interpretability of supervised probes (Section 4.2), SAE's sparsity-driven proxy space (Section 4.4), and output-oriented investigation (Section 4.5) of conventional logit-based probing approaches. VSA-based probing is particularly effective in experimental settings that require a constrained or structured conceptual feature space, a limitation of SAE-based analysis (Section 5.4), while yielding deeper and less variable findings than token–based analysis (Section 5.3). Further, our ingestion algorithm (Section 4.3) bypasses layer selection of conventional layer-wise probing approaches.

Our method enables joint input–output feature extraction that uncovers non-trivial insights into the semantic information encoded within varied language models:

- Section 5 unveils *differences in concept-based compositionality among diverse models and input types* (RQ1–RQ5). For instance, Section 5.2 shows that OLMo-2 possesses a more contextually rich latent structure compared to GPT-2. This also identifies more frequent blank representations tied to specific input domains, which may be tied to the specific type of reasoning required.

- Section 6 studies concept-based dynamics before and after text generation during QA inference across diverse language models (RQ6). The analysis suggests distinct sources of failure: Llama's incorrect answers appear to be associated with a loss of focus on question-related concepts, whereas GPT-2's errors are associated with a shallower representation of the QA task.

Lastly, VSA-based probing applies to any autoregressive language model, is compatible with all Hugging Face models, and introduces a lightweight, layer-agnostic probe (Appendix Q). Moreover, combining VSA generality with hypervector algebra provides a promising way to steer model's generation, study concept-oriented dynamics in other NLP tasks (Appendix I), and unveiling multi-modal latent features (Appendix P).

**Limitations.** The main limitation of this work (see also Appendix A) is its reliance on unidirectional transformation (Equation 3), preventing causal validation in its current implementation. Although intervention-based analysis is outside our primary scope, Appendix I.4 presents experiments using a bidirectional design to steer text generation using a VSA-based methodology.

Distinguishing probe-induced learning from genuine information decoding remains an ongoing challenge in interpretability research (e.g., Belinkov, 2022), as the human-interpretable information encoded in LLM embeddings is not explicitly known. To mitigate these confounding effects, this work employs several strategies. These range from evaluating on syntactically diverse test inputs ($\bar{\mathcal{S}}$, $\bar{\mathcal{Q}}$) distinct from the training data ($\mathcal{S}$, $\mathcal{Q}$) across multiple experimental settings (Sections 5 and 6), to defining an indirect learning objective that targets a proxy space rather than the direct task-specific features typical of standard supervised probes. While we cannot directly quantify the effectiveness of these strategies, Appendix F.1 presents two control tests that assess whether the probe genuinely decodes information from LLM representations, a null model with randomized input embeddings and probing the learned proxy space with concepts unrelated to inputs.

While the data-agnostic design of VSAs removes any reliance on the LLM's output vocabulary, VSA-based probing still requires a predefined set of latent features. Nonetheless, this set functions as a vocabulary free of constraints on symbol count, type, or origin. It serves as a comprehensive repository of all possible latent features. Appendix I highlights the broader applicability of a VSA-based methodology, including the creation of VSA-based representations for diverse input types and downstream tasks, as well as steering the text generation process of language models.

## Reproducibility statement

The submission includes the source code and the synthetic corpora, which will be released upon acceptance. A *README.md* file provides instructions for reproducing the methodology and the experimental results.

Section 4 outlines the full pipeline, from data creation (Section 4.1, Section 4.2) to model training (Section 4.3, Section 4.4). Additional training details appear in Appendix D.1, and the model architecture in Appendix C. The ingestion algorithm is further detailed in Appendix B. Lastly, Appendix D.2 lists the Hugging Face links for all LLMs, and Appendix Q reports computational costs.

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

## Appendix contents

## A    Limitations

While we apply data augmentation and test on syntactically different inputs to mitigate confounding on information decoding (Section 4.1 and Section 5.1), we could not measure the effectiveness of these strategies.

Table 1, Table 3 and Table 10 report on the actual concepts identified by our probing method. The label `Key values` denotes instances where the probe retrieves a key of an analogy pair with a concept linked to it in a different domain (see `Australia` in Section 4.1). This outcome can be viewed as an artifact of our probe, revealing the confounding influence of memorized key-value associations. Nevertheless, such cases constitute only a small fraction, 2% of the 114,099 textual inputs processed across all models, covering *Key / Key Values*, *Example Value / Key Values*, and *Key Values / Target* shown in Table 1. To further investigate potential confounding effects from probe learning, we introduce two control tests, as proposed in (Hewitt & Liang, 2019). Using randomly-permuted input embeddings ($e_s$) as a null model (Gotelli et al., 2012), and applying the unbinding operation on VSA encodings ($y_s$) with concept pairs unrelated to inputs, respectively, *permuted* and *unrelated baseline* in Appendix F.

While our approach avoids dependence on the LLM's vocabulary of DLA-based methods (Section 2) due to the data-agnostic nature of VSAs, it still requires a predefined set of concepts. This set can however be seen as an vocabulary with no practical constraints on the cardinality, type and source of its symbols.

## B    Algorithm to process neural embeddings of LLMs

---

**Algorithm 1:** Ingestion procedure $F$

---

**Data:** Textual sequence $s \in \mathcal{S}$
**Result:** Compressed model state for its next token prediction
**begin**

    // Get the residual stream from the language model
    $\mathbf{H} \leftarrow \text{LLM}(s) \in \mathbb{R}^{L \times T \times d}$ ;

    // Retain embeddings of the last token from the bottom half of the layers
    $\mathbf{H}^\star \leftarrow \mathbf{H}[L/2 : L, -1]$;

    // Apply K-Means clustering
    $\mathbf{C} \leftarrow \text{KMeans}_{K=5}(\mathbf{H}^\star) \in \mathbb{R}^{K \times d}$ ;

    // Sum pooling across the centroids
    $\mathbf{e}_s \leftarrow \sum_{k=1}^{5} \mathbf{C}_k \in \mathbb{R}^d$ ;

    **return** $\mathbf{e}_s$

---

## C   Architecture of our *Hyperdimensional probe*

Table 5: Configuration of the neural VSA encoder $\mathcal{M}$ for an input embedding dimension equal to $d$.

| Component | Input Dim | Output Dim | Note |
|---|---|---|---|
| **Input Layer** | | | |
| Linear Layer | $d$ | 4096 | - |
| Normalization | - | - | LayerNorm (4096) |
| Activation | - | - | GELU |
| **Residual Block 1** | | | |
| Linear Layer | 4096 | 4096 | GELU activation |
| Normalization | - | - | LayerNorm (4096) |
| Dropout | - | - | $p = 0.5$ |
| Residual Connection | - | - | Identity |
| **Residual Block 2** | | | |
| Linear Layer | 4096 | 4096 | GELU activation |
| Normalization | - | - | LayerNorm (4096) |
| Dropout | - | - | $p = 0.5$ |
| Residual Connection | - | - | Identity |
| **Output Layer** | | | |
| Normalization | - | - | LayerNorm (4096) |
| Linear Layer | 4096 | 4096 | - |
| Activation | - | - | Tanh |

Trainable parameters with: $d = 1024, 55M$
$d = 2048, 59M$
$d = 4096, 67M$
$d = 5120, 71M$

# D  Training performance of the neural VSA encoders

Table 6: Training performance of our neural VSA encoder $\mathcal{M}$ on the test set. Order by model size. `BASELINE` introduces a comparison against a null model with randomly-permuted input embeddings.

| Large Language Model | | | | Cosine similarity | Binary accuracy |
|---|---|---|---|---|---|
| **Name** | **Parameters** | **Embedding dimension** | **Layers from residual stream** | | |
| `BASELINE` | - | 5120 | - | 0.500 | 0.750 |
| Llama 4, Scout, 17B-16E | 109 B | 5120 | $24^{th}$ to $48^{th}$ \|25\| | 0.890 | 0.934 |
| OLMo-2 | 32 B | 5120 | $32^{nd}$ to $64^{th}$ \|33\| | 0.878 | 0.926 |
| Phi 4 | 14 B | 5120 | $20^{th}$ to $40^{th}$ \|21\| | 0.881 | 0.930 |
| Llama 3.1-8B | 8 B | 4096 | $16^{th}$ to $32^{nd}$ \|17\| | **0.892** | **0.937** |
| Pythia-1.4b | 1.4 B | 2048 | $12^{th}$ to $24^{th}$ \|13\| | 0.861 | 0.916 |
| GPT-2, medium | 355 M | 1024 | $12^{th}$ to $24^{th}$ \|13\| | 0.865 | 0.920 |
| | | | **AVERAGE** | 0.878 ± 0.01 | 0.927 ± 0.01 |

## D.1  Training details

The neural VSA encoder $\mathcal{M}$ was trained for 421 epochs on average via PyTorch Lighting,[8] using early stopping (patient set at 100 epochs) and a batch size of 32. The optimal learning rate was automatically determined using the learning rate finder provided by the aforementioned library, and was approximately set to $3e^{-5}$ on average. We use *AdamW* as the optimizer (weight decay of $1e^{-4}$), applying a learning rate schedule based on Cosine Annealing with Warm Restarts, starting from the 100th epoch and doubling the restart period thereafter. To adapt the batch size after LR restarts, we employed a Gradient Accumulation Scheduler: the effective batch size was doubled at the 110th epoch, quadrupled at the 310th, and increased eightfold at the 410th epoch. During training, the model's outputs are dynamically binarized using the sigmoid function to ensure compatibility with the loss function (Section 4.4). This approach demonstrated better empirical performance than linear min-max normalization.

## D.2  Hugging Face repositories for the considered LLMs

1. Meta AI's **Llama 4, Scout**, huggingface.co/meta-llama/Llama-4-Scout-17B-16E

2. Meta AI's **Llama 3.1**, huggingface.co/meta-llama/Llama-3.1-8B

3. Microsoft's **Phi-4**, huggingface.co/microsoft/phi-4

4. EleutherAI's **Pythia**, huggingface.co/eleutherai/pythia-1.4b

5. AllenAI's **OLMo-2**, huggingface.co/allenai/OLMo-2-0325-32B

6. OpenAI's **GPT-2**, huggingface.co/openai-community/gpt2-medium

---

[8]lightning.ai/docs/pytorch/stable

## E  Unbinding stage from Section 4.5

Table 7: **Unbinding stage**: Proportions of the best unbinding concepts used for extracting concepts from VSA encodings across different models, with overall mean and standard deviation. `Key` refers to cases where the candidate concept corresponds to the key of the target pair ($b_1$), while `NONE` indicates that no unbinding operations were applied to the probed VSA encoding. `Example` denotes a concept where the key ($a_1$) and value ($a_2$) from the in-context example were pre-bound. Lastly, `Context` represents a scenario where the in-context example ($a_1, a_2$) was pre-bound together with the key of the target pair ($b_1$). On the other hand, `Greedy` means using a concept candidate from the vocabulary, rather than picking it among those of the input. The table has been trimmed to highlight the relevant and common items across the models. We consider the first four strategies to be the most relevant, as they account for 97% of all unbinding operations across models.

| Concept for unbinding (%) | GPT-2 | Pythia | Llama 4, Scout | OLMo-2 | Phi-4 | Llama 3.1 | AVERAGE |
|---|---|---|---|---|---|---|---|
| Key | 65.9 | 74.4 | 83.2 | 83.2 | 87.4 | **87.9** | $80.3 \pm 7.8$ |
| NONE | **22.0** | 16.9 | 11.6 | 8.6 | 7.7 | 7.0 | $12.3 \pm 5.4$ |
| Example Key | **6.0** | 2.6 | 1.0 | 2.1 | 1.5 | 1.7 | $2.5 \pm 1.7$ |
| Context | 1.2 | 2.0 | 2.6 | **4.5** | 1.3 | 1.5 | $2.2 \pm 1.2$ |
| Greedy | 2.1 | 1.9 | 1.3 | 0.9 | 1.2 | 0.9 | $1.4 \pm 0.5$ |
| Example Value | 1.6 | 1.5 | 1.0 | 0.4 | 0.7 | 0.5 | $0.9 \pm 0.5$ |
| Cleaned Example Key | 0.2 | 0.5 | 0.0 | 0.1 | 0.2 | 0.1 | $0.2 \pm 0.2$ |
| Cleaned Example Value | 0.9 | 0.1 | 0.0 | 0.1 | 0.1 | 0.1 | $0.2 \pm 0.3$ |
| Cleaned Key | 0.0 | 0.0 | 0.0 | 0.0 | 0.0 | 0.1 | $0.0 \pm 0.0$ |
| Cleaned Original | 0.0 | 0.0 | 0.0 | 0.0 | 0.0 | 0.0 | $0.0 \pm 0.0$ |
| Example | 0.0 | 0.0 | 0.0 | 0.0 | 0.0 | 0.0 | $0.0 \pm 0.0$ |
| Example Value & Key | 0.0 | 0.0 | 0.0 | 0.0 | 0.0 | 0.0 | $0.0 \pm 0.0$ |
| Example Key & Key | 0.0 | 0.0 | 0.0 | 0.0 | 0.0 | 0.0 | $0.0 \pm 0.0$ |

## F  Experimental results

Table 8: Experimental results on the LLM's analogy-style completion tasks, along with our probing method for retrieving the target concept from model's internal state. The model are ordered based on precision@1 for next-token prediction. Statistical variability is reported using 95% confidence intervals, which more appropriately capture variability for metrics bounded within the [0,1] range. To control for randomness, we also introduce two control tests using Llama 3.1-8B: a comparison against a null model with randomly-permuted input embeddings ($e_s$, *permuted baseline*), and extraction of concept pairs unrelated to inputs ($y_s$, *unrelated baseline*).

| MODEL | LLM's Next Token | | DLA-based Probing | SAE-based Probing | | VSA-based Probing | |
|---|---|---|---|---|---|---|---|
| | Precision@1 | Precision@5 | Precision@1 | Precision@10 | Precision@100 | Precision@1 | Precision@5 |
| *Permuted baseline* | - | - | - | - | - | 0.080 (079-082) | 0.103 (101-104) |
| *Unrelated baseline* | - | - | - | - | - | 0.099 (097-101) | 0.105 (103-107) |
| Llama 4 Scout, 17B-16E | 0.081 (079-082) | 0.501 (498-504) | **0.773** (771-775) | 0.604 (601-607) | 0.923 (921-924) | 0.866 (864-868) | 0.875 (873-877) |
| GPT-2, medium | 0.267 (265-270) | 0.548 (545-551) | 0.300 (298-303) | 0.063 (062-064) | 0.402 (399-404) | 0.692 (689-695) | 0.702 (699-704) |
| Pythia-1.4b | 0.369 (366-372) | 0.654 (651-657) | 0.413 (410-416) | 0.205 (203-208) | 0.863 (861-865) | 0.778 (776-781) | 0.790 (788-792) |
| Llama 3.1-8B | 0.352 (349-354) | 0.546 (543-549) | 0.467 (464-470) | 0.595 (592-598) | **0.926** (924-927) | **0.891** (889-893) | **0.908** (907-910) |
| Phi 4 | 0.519 (516-522) | **0.753** (750-755) | 0.585 (582-588) | **0.749** (746-751) | 0.918 (916-919) | 0.887 (886-889) | 0.904 (902-905) |
| OLMo-2 | **0.529** (526-532) | 0.721 (719-724) | 0.714 (712-717) | 0.526 (523-529) | 0.893 (891-895) | 0.879 (877-881) | 0.892 (890-894) |
| AVERAGE | 0.352 | 0.621 | 0.542 | 0.457 | 0.821 | 0.832 | 0.845 |

### F.1  Validation strategy

To assess the effectiveness of our probe, we conduct two control tests (see Table 8) as proposed in *"Designing and interpreting probes with control tasks"* by Hewitt & Liang (2019):

1. **Permuted Baseline**: We compared our outputs against a null model by inputting the trained probe with randomly permuted LLM embeddings;

2. **Unrelated Baseline**: We attempt to extract concepts that are unrelated to the input using VSA-based probing.

Both tests yielded very low precision probing scores, reinforcing the effectiveness of our method. These results show that:

- Applying our VSA-based probing (see Equation 4) using concepts irrelevant to input texts results in meaningless outputs;

- Corrupted or nonsensical input embeddings also produce poor results.

That said, it is crucial to recognize a fundamental limitation of all probing approaches: by definition, the human-interpretable information encoded in LLM embeddings is not explicitly known. Consequently, no probing method can provide absolute certainty in decoding such information. To address this, we further validated our method by evaluating the trained probes on textual inputs distinct from those used during training, thereby reinforcing the reliability of our information decoding approach.

### F.2 Diagnosing erroneous answers from Llama 4

Llama 4 most frequently generated a white space token for our corpus $\bar{\mathcal{S}}$, accounting for 76% of its outputs, considerably higher than the 8% average observed in the other models (30% for Llama 3.1). Its next most common tokens were: `?` (9%), `what` (6%) and `x` (0.7%). The target token had a median rank of 5, with its SoftMax score trailing the top-1 token by a median difference of 0.85 (Appendix L.1), which starkly contrasts other models with 0.05. Thus, the model confidently predicted a space, with the target word often within its top five predictions. These insights, and the strong performance of our hyperdimensinal probe (probing@1 = 87%), suggest issues in handling the syntactical structure of our corpus rather than lack of analogical reasoning. Possibly influenced by its tokenizer (see space-token frequency in the other Llama), which emphasizes prompt engineering importance and variability caused by models' tokenizers. This may be further worsened by the model's multimodality and the complexity of its MoE architecture.

## G Experimental comparison

**Logit Attribution.** Initially, Section 5.3 compares our VSA-based results to those generated by the technique of Direct Logit Attribution (DLA; Section 2); as it requires no extra steps such as feature-naming. This makes DLA the most direct and unambiguous comparison for our experimental setting. It outputs a single, unique and unambiguous feature (token) constrained by the model's output vocabulary, enabling a direct comparison through a fuzzy token-to-concept matching with our concept set.

**Sparse Autoencoders.** Conversely, while our controlled vector space, defined by VSA encodings, parallels the SAE proxy layer, our methodology adopts a top-down strategy by querying this proxy space with predefined concepts (Equation 4), whereas SAEs follow a bottom-up process that assigns labels to all activated features post hoc. This bottom-up paradigm uncovers an unbounded set of latent features without inherent relevance filtering, necessitating exhaustive feature naming and additional selection to isolate those aligned with our bounded input–output concept framework. The manual intervention required in SAE-based methods, from feature naming to filtering, prevents them from being directly and unambiguously comparable to our supervised approach. Section 5.4 presents an analysis based on Sparse Autoencoders (SAEs; Section 2), detailing the experimental choices undertaken to enable a comparison with our VSA-based findings. Although Section 2 discuss the fundamental methodological differences between SAEs and our VSA-based approach, most notably the bottom-up, unsupervised discovery of latent features by SAEs, we designed our experimental protocol to provide a comparison that is as fair as possible while respecting the experimental setup underlying our work and the typical operating regime of SAE-based analyses.

The objective of this comparison is to assess how the two methodologies behave within the same experimental setting. Our results indicate that, in bounded conceptual frameworks requiring a predefined and constrained feature space, SAE-based analyses yield concept-oriented findings that are less structured and exhibit greater noise than those obtained with our VSA-based approach. We attribute these differences to the distinct operating principles of the two methods, particularly the bottom-up, unsupervised feature discovery employed by SAEs. Accordingly, the experimental protocol in Section 5.4 was designed to provide a comparison that is as comprehensive and fair as possible while remaining faithful to both the assumptions underlying our methodology and the standard operating regime of SAE-based analyses. Alternative SAE configurations, such as extracting features via input clustering using an LLM-as-Judge strategy instead of DLA, would certainly constitute an interesting extension. However, they would not fundamentally alter the methodological differences between the two approaches, nor the central research question addressed by our comparison. In particular, the comparison is intended to evaluate SAE-based concepts in a constrained conceptual feature space, where the bottom-up feature learning paradigm of SAEs remains unchanged regardless of the specific feature extraction strategy. Moreover, introducing additional SAE variants would substantially increase the complexity of the experimental study, making direct comparisons with our proposed methodology less transparent while expanding the scope of the paper beyond its primary objectives.

**Supervised probes.** Our VSA-based encoder does qualify as a supervised probe, as it is trained to map LLM internal representations (i.e., residual stream) into interpretable, human-understandable features (i.e., VSA encodings). Supervised probes are typically designed for specific experimental goals or target features, ranging from syntactic structure, as in *"A Polar Coordinate System Represents Syntax in Large Language Models"* (Diego Simon et al., 2024); to real-world knowledge, as in *"Language Models Represent Space and Time"* (Gurnee & Tegmark, 2023); and to abstract semantics, as in *"The Geometry of Truth"* (Marks & Tegmark, 2023). However, our probe is specifically designed around VSAs principles, so direct comparisons with non-VSA probes would require fundamentally different approaches, not grounded in VSAs.

### G.1 DLA-based experimental results

To validate our results, we apply DLA to all models using $\bar{S}$, as it allows direct baseline without extra steps such as feature naming or filtering required in SAE analysis. See Appendix G for details.

We adopt simple, fuzzy token-to-concept matching approach with our concept set (e.g., pes $\mapsto$ peso), and consider projected next-token predictions (Appendix G.2) from the model's middle to last layers of the

last token, as VSA probing. DLA produces no concepts in nearly 30% of analogies on average (see `NONE` in Table 9; +17% compared to VSA, Table 1), while yielding the target with its key in 26% of the cases (-50%). In instances without concepts from DLA, our VSA-based probe extracts, on average, the key-target pair in 57% of all analogies (Table 3), while returning none for 28%. For instance, for the analogy `king is to queen as son is to` $\mapsto$ `daughter`, using OLMo-2, our probe extracts the key-target concepts (`son` and `daughter`), while DLA produces no concepts. The model predicts the next token prediction as `?` with a softmax score of 0.06, followed by `father` (0.05); the target word has a rank of 37. Focusing on next-token representations, and thus capturing surface-level features, DLA exhibits inferior probing capabilities compared to ours, which compromise subsequent interpretability analyses of LLM embeddings. On the other hand, we observe substantial variance within this subset during VSA probing. Across models (Table 3), our probe fails to retrieve any concepts in 43% of cases for Llama 4, compared to only 14% for Llama 3.1. GPT-2 confirms greater representativeness for the in-context example. There is also variation across analogy categories in this subset (Table 10): for OLMo-2, linguistic analogies show the highest retrieval rates for `Context | Target` (7.4% and 4.4%), whereas mathematical analogies shows nearly no concept retrieval (91%), confirming common blank representations. Appendix G.3 shows that, in cases where VSA fails, also DLA frequently yields no concepts rather than other relevant concepts.

Table 9: Concepts extracted using the DLA probing technique on the full corpus $\bar{\mathcal{S}}$ with all LLMs. Likewise in our VSA-based probing, we focus on the same middle-to-bottom range of model's hidden layers of the last token. The table highlights the key common items across models, with the first six cases covering over 95% of all extracted concepts.

| Extracted Concepts (docs, %) | GPT-2 | Pythia | Llama4, Scout | OLMo-2 | Phi-4 | Llama 3.1 | AVERAGE | Δ VSA |
|---|---|---|---|---|---|---|---|---|
| NONE | 33.9 | 32.8 | 15.4 | 14.6 | 33.1 | **47.4** | 29.5 ± 11.4 | +17.3 |
| **Target** | 15.0 | 18.0 | **36.7** | 29.0 | 34.4 | 22.5 | 25.9 ± 8.1 | +25.8 |
| Key \| **Target** | 12.6 | 19.3 | 38.4 | **38.5** | 22.7 | 22.1 | 25.6 ± 9.7 | - 50.4 |
| Key | 9.7 | **10.4** | 6.3 | **10.4** | 6.0 | 4.5 | 7.9 ± 2.4 | +5.0 |
| Example Value | **12.8** | 5.7 | 0.3 | 0.7 | 0.9 | 0.5 | 3.5 ± 4.6 | +3.5 |
| Example | **9.0** | 5.3 | 0.3 | 1.7 | 1.0 | 0.5 | 3.0 ± 3.2 | +0.7 |
| Example Value \| **Target** | 1.0 | **2.1** | 0.7 | 0.6 | 0.3 | 0.9 | 0.9 ± 0.6 | +0.9 |
| Example Key | **3.0** | 1.5 | 0.1 | 0.2 | 0.1 | 0.1 | 0.8 ± 1.1 | +0.7 |
| Context \| **Target** | 0.6 | 0.3 | 0.5 | **1.5** | 0.3 | 0.4 | 0.6 ± 0.4 | -1.5 |

Table 10: Percentages of extracted factors by analogy category considering the subset of instances when the DLA yields no concept for OLMo-2.

| Extracted concepts (docs, %) | Morphological Modifiers | Verbal & Grammatical Forms | Factual Knowledge | Semantic Relations | Mathematics | Semantic Hierarchies | AVERAGE |
|---|---|---|---|---|---|---|---|
| Key \| Target | **90.3** | 83.4 | 70.1 | 79.4 | 0.0 | 41.5 | 60.8 ± 30.3 |
| NONE | 1.6 | 2.7 | 14.3 | 1.7 | **91.1** | 21.1 | 22.1 ± 31.1 |
| Example | 0.7 | 0.7 | 4.7 | 8.5 | 0.0 | **15.0** | 4.9 ± 5.1 |
| Key | 1.6 | 2.7 | 3.8 | 1.4 | 0.0 | **5.1** | 2.4 ± 1.7 |
| Key \| Pair Values | 1.3 | 0.9 | 0.0 | 5.1 | 0.0 | **11.6** | 3.2 ± 4.2 |
| Context \| Target | 4.4 | **7.4** | 0.8 | 1.6 | 0.0 | 0.6 | 2.5 ± 2.6 |
| Out-of-Context | 0.2 | 0.6 | 1.3 | 0.0 | **8.8** | 0.9 | 1.9 ± 2.9 |
| Context | 0.0 | **0.3** | 0.1 | 0.0 | 0.0 | **0.3** | 0.1 ± 0.1 |

## G.2 Raw results obtained though the DLA probing technique

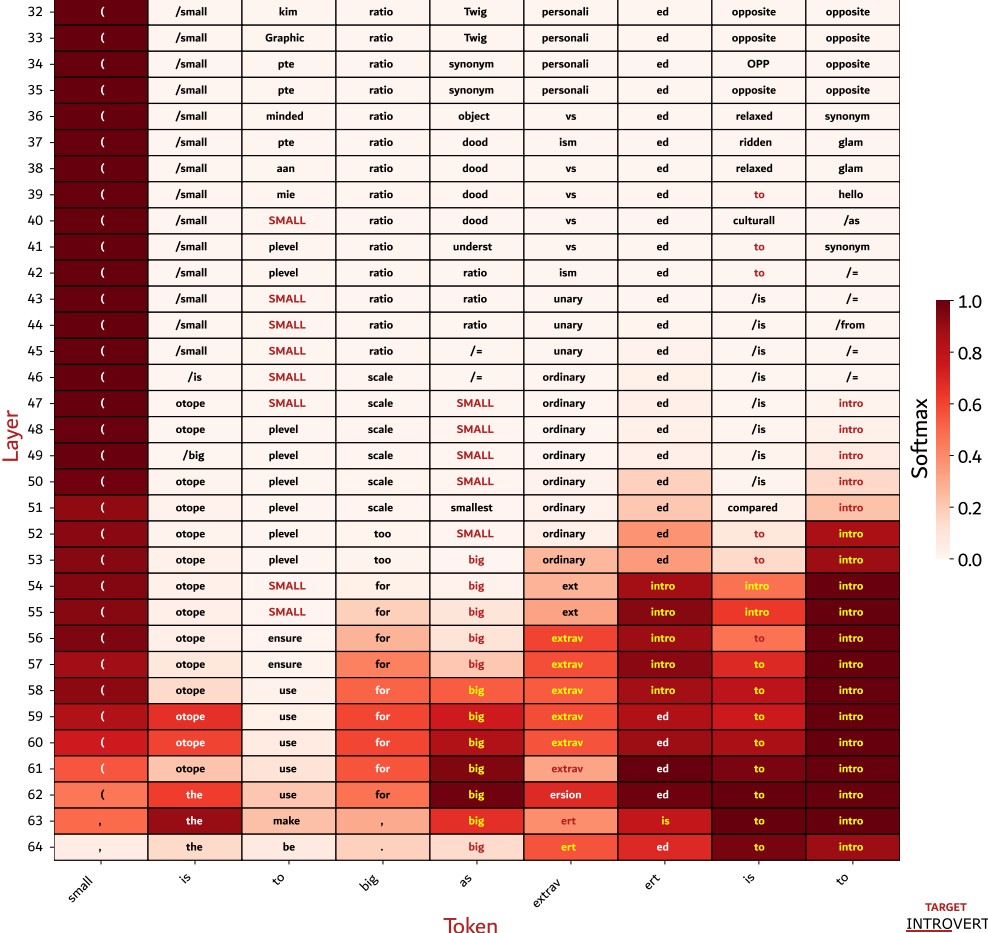

Figure 6: Comprehensive raw outputs obtained though DLA on OLMo-2 for a sampled analogy.

### G.3 Concepts extracted by DLA when VSA yields no concepts

Table 11: Concepts extracted though DLA-based probing when VSA yields no concepts. DLA extract no concepts in the majority of the instances (59 ± 15 %), highlighting high variability among models.

| Extracted concepts (docs,%) | GPT-2 | Pythia | Llama3 | Phi-4 | OLMo-2 | Llama4 | AVERAGE |
|---|---|---|---|---|---|---|---|
| None | 40.9 | 46.5 | 83.1 | 72.7 | 55.9 | 53.9 | 58.8 ± 14.8 |
| **Target** | 8.1 | 9.0 | 5.3 | 15.7 | 14.3 | 23.3 | 12.6 ± 6.4 |
| Key | 7.5 | 11.7 | 3.4 | 4.3 | 13.2 | 9.1 | 8.2 ± 3.7 |
| Key \| **Target** | 6.9 | 8.7 | 5.7 | 3.6 | 11.8 | 11.9 | 8.1 ± 3.3 |
| Example Value | 17.9 | 11.1 | 1.3 | 1.0 | 1.3 | 0.6 | 5.5 ± 6.7 |
| Example | 12.1 | 6.6 | 0.4 | 2.2 | 1.7 | 0.1 | 3.9 ± 4.3 |
| Example Key | 3.3 | 2.7 | 0.1 | 0.1 | 0.2 | 0.2 | 1.1 ± 1.2 |
| Example Value \| **Target** | 1.0 | 1.7 | 0.1 | 0.0 | 0.2 | 0.2 | 0.5 ± 0.7 |
| Example Value \| Key | 1.0 | 0.6 | 0.1 | 0.0 | 0.4 | 0.3 | 0.4 ± 0.4 |
| Example Key \| Key | 0.3 | 0.4 | 0.1 | 0.0 | 0.1 | 0.1 | 0.2 ± 0.2 |
| Example Value \| Key \| **Target** | 0.3 | 0.3 | 0.0 | 0.0 | 0.3 | 0.2 | 0.2 ± 0.1 |
| Context \| Target | 0.3 | 0.2 | 0.3 | 0.2 | 0.4 | 0.1 | 0.3 ± 0.1 |
| Example Key \| Target | 0.2 | 0.1 | 0.0 | 0.0 | 0.0 | 0.0 | 0.1 ± 0.1 |
| Target \| Example | 0.1 | 0.1 | 0.0 | 0.0 | 0.1 | 0.0 | 0.1 ± 0.1 |

### G.4 Training performance of Sparse Autoencoders

Table 12: Training performance of SAEs on the test set, order by model size. BASELINE compares against a null model using randomly permuted input embeddings.

| Large Language Model | | | Sparse Autoencoder (SAEs) | | Cosine similarity |
|---|---|---|---|---|---|
| Name | Layers from residual stream | Embedding dimension | Latent dimension | Top-k | |
| BASELINE | - | 5120 | 20480 | 410 | 0.584 |
| Llama 4, Scout, 17B-16E | $24^{th}$ to $48^{th}$ \|25\| | 5120 | 20480 | 410 | 0.979 |
| OLMo-2 | $32^{nd}$ to $64^{th}$ \|33\| | 5120 | 20480 | 410 | 0.976 |
| Phi 4 | $20^{th}$ to $40^{th}$ \|21\| | 5120 | 20480 | 410 | 0.985 |
| Llama 3.1-8B | $16^{th}$ to $32^{nd}$ \|17\| | 4096 | 16384 | 328 | 0.979 |
| Pythia-1.4b | $12^{th}$ to $24^{th}$ \|13\| | 2048 | 8192 | 164 | 0.994 |
| GPT-2, medium | $12^{th}$ to $24^{th}$ \|13\| | 1024 | 4096 | 82 | 0.988 |
| | | | | **AVERAGE** | 0.984 ± 0.01 |

# H Cosine similarities among the items of the VSA codebook

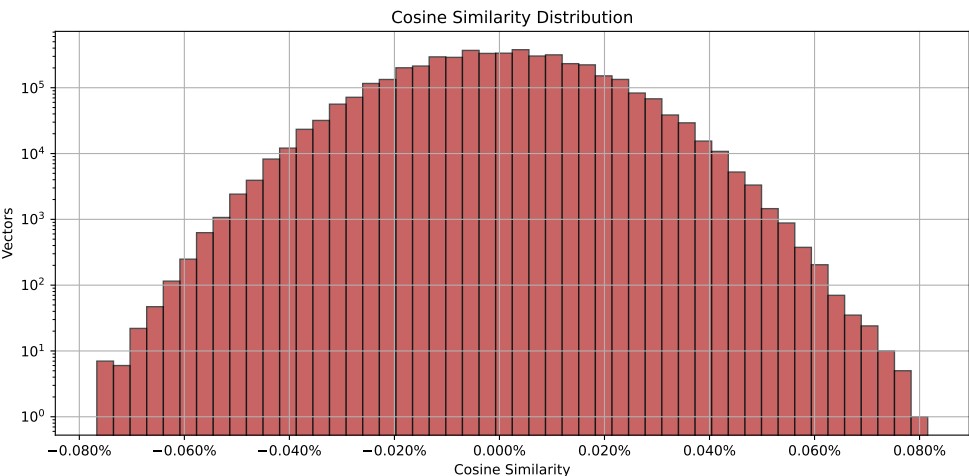

Figure 7: Distribution of pair-wise cosine similarities among the items of the codebook.

# I Applicability to other domains

## I.1 Generalization of input representation

VSA representations are automatically generated from input features, with their construction guided by the probing objective and the target latent features. While our work focuses on textual inputs with well-defined semantics, allowing straightforward extraction of input features (i.e., words), the underlying principle is flexible and generalizable. Equation 2 illustrates the creation of input representations via binding and bundling operations for our specific input template and downstream task. The hyperdimensional algebra underlying VSA allows this approach to generalize to other textual formats, NLP tasks, and even multi-modal data (see appendix P).

Scalability challenges depend largely on the nature of the input features. For tasks such as toxicity detection, expert-labeled data or specialized feature extraction pipelines may be required. For example, mapping the phrase *"You are a pathetic excuse for a human just like the rest of your kind"* to a conceptual form such as $(\phi_{\text{attack}} \odot \phi_{\text{insult}}) + (\phi_{\text{attack}} \odot \phi_{\text{identity}})$ requires human expertise. Once features are extracted, however, constructing VSA encodings is automatic, efficient, and scalable. VSA probing can then uncover encoded concepts in the LLM vector space, for instance:

$$y_s \oslash \phi_{\text{attack}} = \phi_{\text{identity}} + \text{noise}$$

In contrast, tasks based on syntactic structures offer more scalable input extraction. For example, the sentence *"The city of Turin is in Italy"* can be processed with conventional techniques such as POS tagging and Semantic Role Labeling (SRL). A VSA encoding can then be automatically created:

$$(\phi_{\text{NOUN}} \odot \phi_{\text{city}}) + (\phi_{\text{PROPN}} \odot \phi_{\text{Turin}}) + (\phi_{\text{VERB}} \odot \phi_{\text{be}}) + (\phi_{\text{PROPN}} \odot \phi_{\text{Italy}})$$

## I.2 Applicability to other downstream tasks

Although we demonstrate VSA-based probing using analogy-competition tasks, the methodology is generalizable to other experimental settings. The analogy-based dataset was chosen to:

- provide a simple, controlled, and interpretable evaluation environment;
- elicit LLMs to focus on concepts and their inherent relationships;
- probe the LLM vector space with inputs spanning a spectrum of reasoning tasks.

Thanks to the flexibility of VSAs and hypervector algebra, VSA-based probing can be applied to a wide variety of experimental settings with different:

1. **Downstream tasks**. Our decoding paradigm can be used for linguistic feature extraction, toxicity detection, or bias classification;

2. **Textual templates**. For example, in question-answering setting, an input text in such as *"Who wrote the play Romeo and Juliet?"* can be encoded as

$$(\phi_{task} \odot \phi_{question}) + (\phi_{relation} \odot \phi_{writtenBy}) + (\phi_{play} \odot \phi_{Romeo\&Juliet})$$

   allowing the VSA to query LLM representations and reveal which concepts are strongly represented or linked to the predicted answer;

3. **Modalities**. As discussed in Appendix P, inputs combining text with other modalities could also be encoded and probed via VSAs.

VSA-based probing thus provides a unified, flexible framework for examining how LLMs encode and relate abstract input features, from syntactic structures to high-level concepts such as gender bias or toxic language.

### I.3 Handling of Out-of-Domain Data

The neural VSA encoder, a central component of the hyperdimensional probe (Section 4.4), is designed to map concepts with their corresponding neural representations, specifically, to map VSA encodings onto the appropriate directions or subspaces within LLM embeddings. To achieve this, the encoder must be exposed to a few examples of the latent features being targeted, whether drawn from in-domain or out-of-domain distributions. In other words, the hyperdimensional probe cannot extract latent features (i.e., concepts) that it has never encountered.

For instance, one might focus on toxicity by training on toxic content from social platforms, thereby mapping textual inputs to toxicity-related concepts (see Appendix I.1). An example could be the phrase "You are a pathetic excuse for a human just like the rest of your kind", which would be associated with the concept. Once trained, the encoder can then be applied to out-of-domain text to identify potential toxicity-related concepts.

Moreover, we evaluate our probe on textual inputs that differ from those used during training (Figure 2). This allows us both to control for confounding effects and to validate the probe's out-of-domain performance. While and share the same underlying semantic concepts, they consist of distinct texts with a different syntactic structure. Consequently, they can be seen as out-of-domain inputs with respect to the LLM embeddings, the inputs processed by our trained encoder.

Importantly, the probe's performance is not affected by the size or construction strategy of the codebook. The codebook, essentially the list of all features, functions such as a vocabulary, unconstrained in terms of cardinality, type, or source of its symbols, thanks to the properties of VSA. However, as with all interpretability probes, its effectiveness ultimately depends on the type of feature being targeted. Consequently, the strategy used to construct input representations (i.e., VSA encodings) directly impacts probe performance. For example, if the probe attempts to map or extract latent features that are unlikely to be encoded in the LLM's vector space, the resulting outputs may be unreliable or meaningless (see the "unrelated baseline" in Appendix F).

### I.4 Applicability for steering text generation

As logical extension, we are working on testing a bidirectional transformation to enable VSA-based steering, shifting from an encoder-only model (Figure 3; Equation 3) to an encoder-decoder architecture:

$$\mathcal{M} : \mathbb{R}^d \to \{-1, 1\}^D \to \mathbb{R}^d$$

We have already obtained promising preliminary results in steering autoregressive text generation. To isolate and manipulate language concepts (i.e., identifying specific directions or subspaces within the LLM space), we train an autoencoder-like model with a combined loss function. This model is designed to reconstruct the LLM embeddings while internally representing the target VSA concept:

$$e(\text{"Die Hauptstadt Frankreichs ist Paris"}) \mapsto \phi_{\text{german}} \mapsto e(\text{"Die Hauptstadt Fran..."}) \tag{6}$$

These training experiments achieved an average test-set binary accuracy of 0.99 for VSA concepts during the encoder stage and a cosine similarity of 0.96 when reconstructing the original LLM embeddings in the decoder stage. Afterwards, to steer the text generation, we project the VSA concepts back into the LLM representation space using the decoder stage (yielding, e.g., $e_{\text{german}}$ and $e_{\text{spanish}}$). We then subtract the source language concept from the hidden representations while adding the target language feature:

$$\hat{h}_s = (h_s - \langle h_s, \hat{v}_{\text{src}}\rangle \hat{v}_{\text{src}}) + \alpha\, \hat{v}_{\text{tgt}}^{\perp} \tag{7}$$

where $\alpha$ is the steering factor, and $\hat{v}_{\text{src}}$, $\hat{v}_{\text{tgt}}$ represent the source and target concept vectors (such as $e_{\text{german}}$ and $e_{\text{spanish}}$). For example, taking the last-token embeddings of the French sentence *"Est situé dans l'état de l'Utah"*, we can successfully steer the model's text generation toward Spanish, producing: *"El Chorro es un pueblo de la comarca de la Costa del Sol"*. Table 13 presents a language accuracy heatmap highlighting these preliminary cross-lingual steering capabilities.

Table 13: Language accuracy heatmap under VSA-based steering.

|        | DE   | EN   | ES   | FR   | IT   | JA   | RU   | ZH   |
|--------|------|------|------|------|------|------|------|------|
| **DE** | 1.00 | 0.61 | 0.39 | 0.40 | 0.70 | 0.38 | 0.47 | 0.70 |
| **EN** | 0.88 | 1.00 | 0.64 | 0.75 | 0.85 | 0.69 | 0.74 | 0.80 |
| **ES** | 0.85 | 0.84 | 1.00 | 0.84 | 0.88 | 0.49 | 0.74 | 0.78 |
| **FR** | 0.77 | 0.63 | 0.50 | 1.00 | 0.80 | 0.45 | 0.56 | 0.67 |
| **IT** | 0.38 | 0.71 | 0.20 | 0.33 | 1.00 | 0.24 | 0.37 | 0.56 |
| **JA** | 0.96 | 0.96 | 0.84 | 0.93 | 0.93 | 1.00 | 0.88 | 0.85 |
| **RU** | 0.80 | 0.63 | 0.65 | 0.73 | 0.75 | 0.36 | 1.00 | 0.82 |
| **ZH** | 0.59 | 0.58 | 0.21 | 0.50 | 0.69 | 0.27 | 0.34 | 1.00 |

## J  Question-answering setting from Section 6

We generate 693,886 training examples $\mathcal{Q}$ from the SQuAD dataset using an augmenting strategy by incrementally considering textual questions with their corresponding features:

$(A_1)$ "What was the `name`" $\mapsto \phi_{\text{name}}$

$(A_2)$ "What was the `name` of the `ship`" $\mapsto \phi_{\text{name}} + \phi_{\text{ship}}$

$(A_3)$ "…" $\mapsto \ldots$

$(A_{n-1})$ "What was the `name` of the `ship` that `Napoleon` sent to `the Black Sea`?" $\mapsto \phi_{\text{name}}$

$\qquad\qquad \mapsto \phi_{\text{name}} + \phi_{\text{ship}} + \phi_{\text{napoleon}} + \phi_{\text{send}} + \phi_{\text{theBlackSea}}$

$(A_n)$ "What was the `name` of the `ship` that `Napoleon` sent to `the Black Sea`?"

`Charlemagne`" $\mapsto (\phi_{\text{name}} + \phi_{\text{ship}} + \phi_{\text{napoleon}} + \phi_{\text{send}} + \phi_{\text{theBlackSea}}) + \phi_{\text{charlemagne}}$

For our experiments, we generate another corpus $\bar{\mathcal{Q}}$ that also includes the contextual text (Wikipedia article) provided for each SQuAD's item:

"Napoleon III responded with a show of force … by the Greek Orthodox Church.

Q: What was the name of the ship that Napoleon sent to the Black Sea? $\qquad\qquad$ (8)

A ($\leq$ 3 words):"

Lastly, we apply our entire pipeline by probing the final state of a language model at the last token (colon) and extracting concepts through comparison with the codebook $\Phi$. We analyze the model's internal state across the text generation process, considering the residual stream at initialization ($\mathbf{H}[\text{seq}_0]$) and after the autoregressive generation of $t$ tokens ($\mathbf{H}[\text{seq}_t]$).

### J.1  Experimental analysis on the QA-based setting with GPT-2

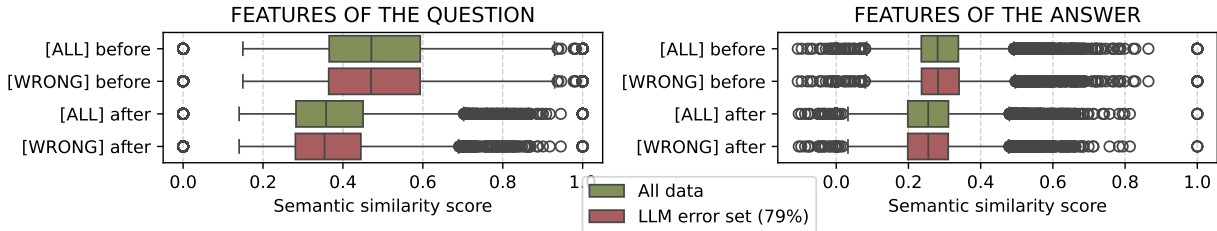

Figure 8: Concepts extracted *before* and *after* the model's text generation, with respect to *question* and *answer* features. Red denotes the subset of failure instances, while green the full sample $\bar{\mathcal{Q}}$.

# K   Spearman correlation for the QA-related experiments

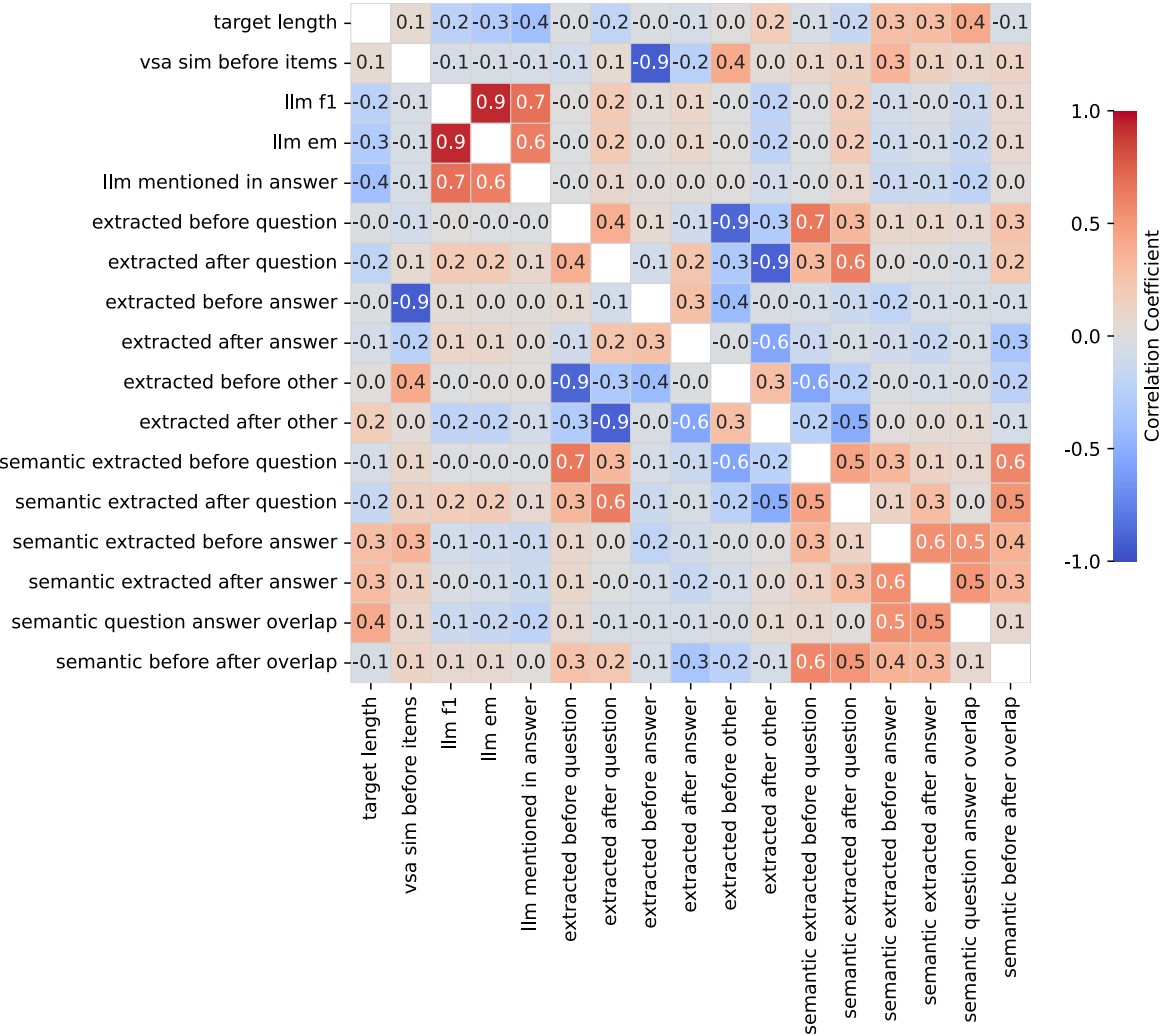

Figure 9: Spearman correlation coefficients computed on $\bar{\mathcal{Q}}$.

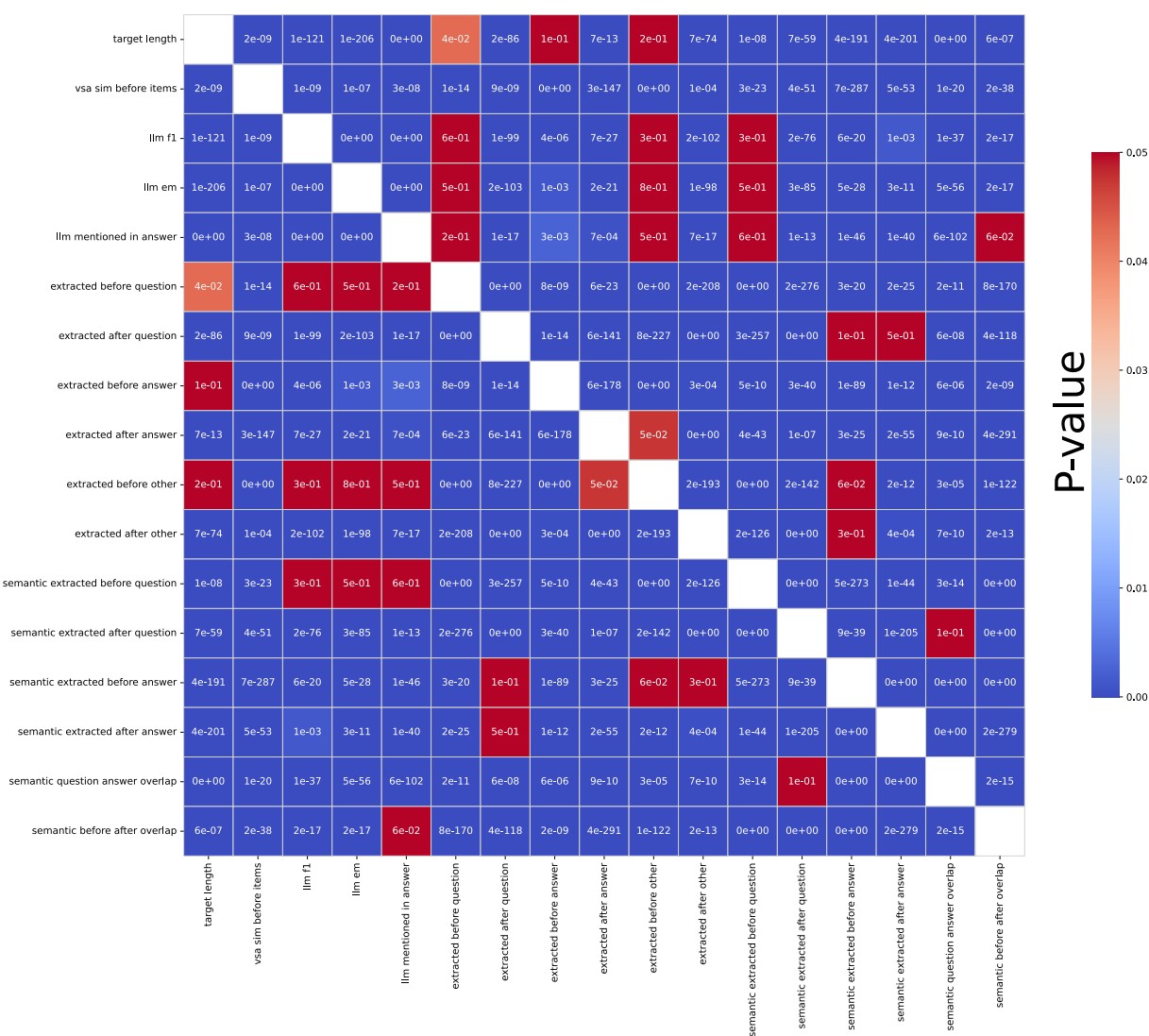

Figure 10: P-values of the Spearman correlation coefficients.

# L    Overview of the experimental metrics

## L.1    Llama 4, Scout

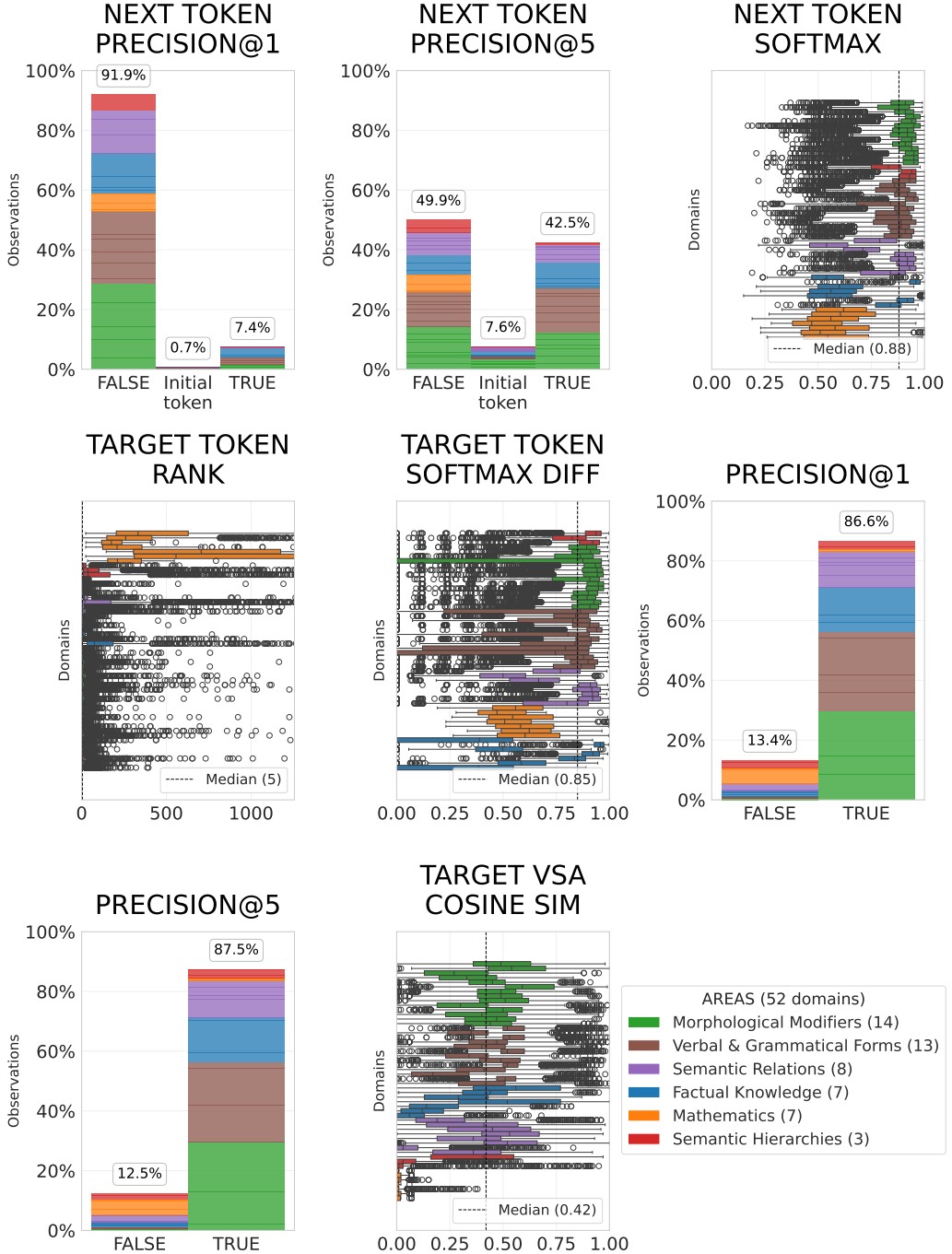

Figure 11: Experimental metrics of the LLM's next-token prediction task and probing performance for Llama 4. *Precision@k* is displayed as a categorical variable, with its binary values portrayed as boolean. The category *initial token* is associated to the special case (0.5) introduced in Section 5.1. We measure VSA noise by computing the cosine similarity between the retrieved target concept and its codebook version Φ.

### L.2    OLMo-2

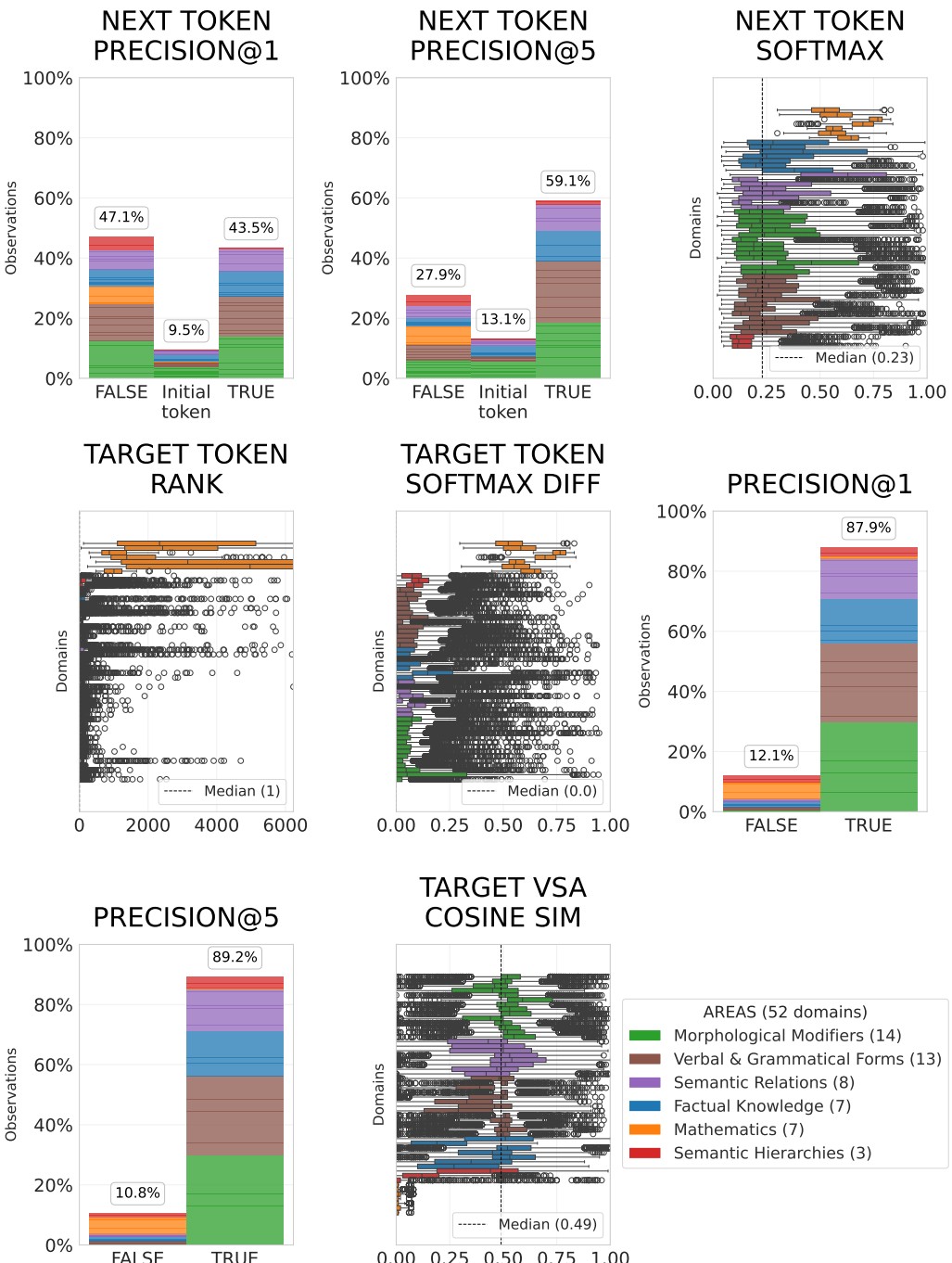

Figure 12: Experimental metrics of the LLM's next-token prediction task and probing performance for OLMo-2. *Precision@k* is displayed as a categorical variable, with its binary values portrayed as boolean. The category *initial token* is associated to the special case (0.5) introduced in Section 5.1.

## M  Synthetic corpus

Table 14: Knowledge bases for our synthetic corpus $\mathcal{S}$.

| Dataset | | Domains | | Sample example |
|---|---|---|---|---|
| Google Analogy Test Set | 12 | capital world, currency, plural, . . . | 33,812 | Denmark : krone = Mexico : peso |
| Bigger Analogy Test Set | 33 | verb+ment, occupation, gender, . . . | 73,471 | queen : king = mother : father |
| Mathematics | 7 | double, square, division2, . . . | 6,816 | 4 : 16 = 5 : 25 |
| | 52 | | 114,099 | |

Table 15: Overview of our experimental set, grouped by tasking an LLM to cluster the domains.

| Category | | Domains | Docs |
|---|---|---|---|
| Morphological Modifiers | 14 | noun+less, adj+ness, . . . | 34,308 (30%) |
| Verbal & Grammatical Forms | 13 | past tense, plural, . . . | 31,219 (27%) |
| Factual Knowledge | 7 | country capital, occupation, . . . | 18,800 (17%) |
| Semantic Relations | 8 | family, genders, . . . | 16,831 (15%) |
| Mathematics | 7 | math double, math division5, . . . | 6,816 (6%) |
| Semantic Hierarchies | 3 | hypernyms, hyponyms, . . . | 6,125 (5%) |
| | 52 | | 114,099 (100%) |

Table 16: All domains, and their corresponding cardinality after data augmentation for training.

| Domain | Examples | Domain | Examples | Domain | Examples |
|---|---|---|---|---|---|
| country_capital | 21801 | capital_world | 18561 | country_language | 12299 |
| antonyms_gradable | 11268 | adj_superlative | 10942 | un+adj_reg | 10614 |
| adj+ly_reg | 10576 | adj_comparative | 10519 | male_female | 10236 |
| noun_plural_reg | 10216 | noun_plural_irreg | 10206 | verb_Ving_Ved | 10164 |
| verb_inf_3pSg | 10112 | animal_sound | 10083 | verb_inf_Ving | 10008 |
| name_nationality | 9998 | verb+er_irreg | 9865 | verb_Ving_3pSg | 9861 |
| verb+able_reg | 9849 | adj+ness_reg | 9849 | animal_shelter | 9833 |
| hypernyms_animals | 9831 | over+adj_reg | 9828 | re+verb_reg | 9821 |
| verb+ment_irreg | 9807 | verb_inf_Ved | 9805 | UK_city_county | 9805 |
| name_occupation | 9801 | noun+less_reg | 9801 | verb_3pSg_Ved | 9801 |
| verb+tion_irreg | 9801 | hypernyms_misc | 9719 | antonyms_binary | 9603 |
| past_tense | 6313 | plural | 4129 | comparative | 3765 |
| present_participle | 3401 | plural_verbs | 3055 | currency | 2983 |
| adjective_to_adverb | 2977 | math_double | 2918 | nationality_adjective | 2818 |
| superlative | 2545 | math_division2 | 2498 | opposite | 2221 |
| math_division5 | 641 | family | 529 | math_squares | 402 |
| math_division10 | 258 | hyponyms_misc | 102 | math_root | 77 |
| math_cubes | 29 | | | | |
| DOMAINS: 52 | | | | TEXTUAL EXAMPLES: 395,944 | |

## N  Declaration of LLM usage

The paper presents a pipeline that treats LLMs as subjects of study, not tools. To enhance interpretability, we adopted an LLM (GPT-4o) to categorize the 52 distinct analogy domains into semantically coherent macro categories (Table 15 in Appendix M).

## O  Dimensionality reduction

### O.1  Average correlations among model's hidden layer

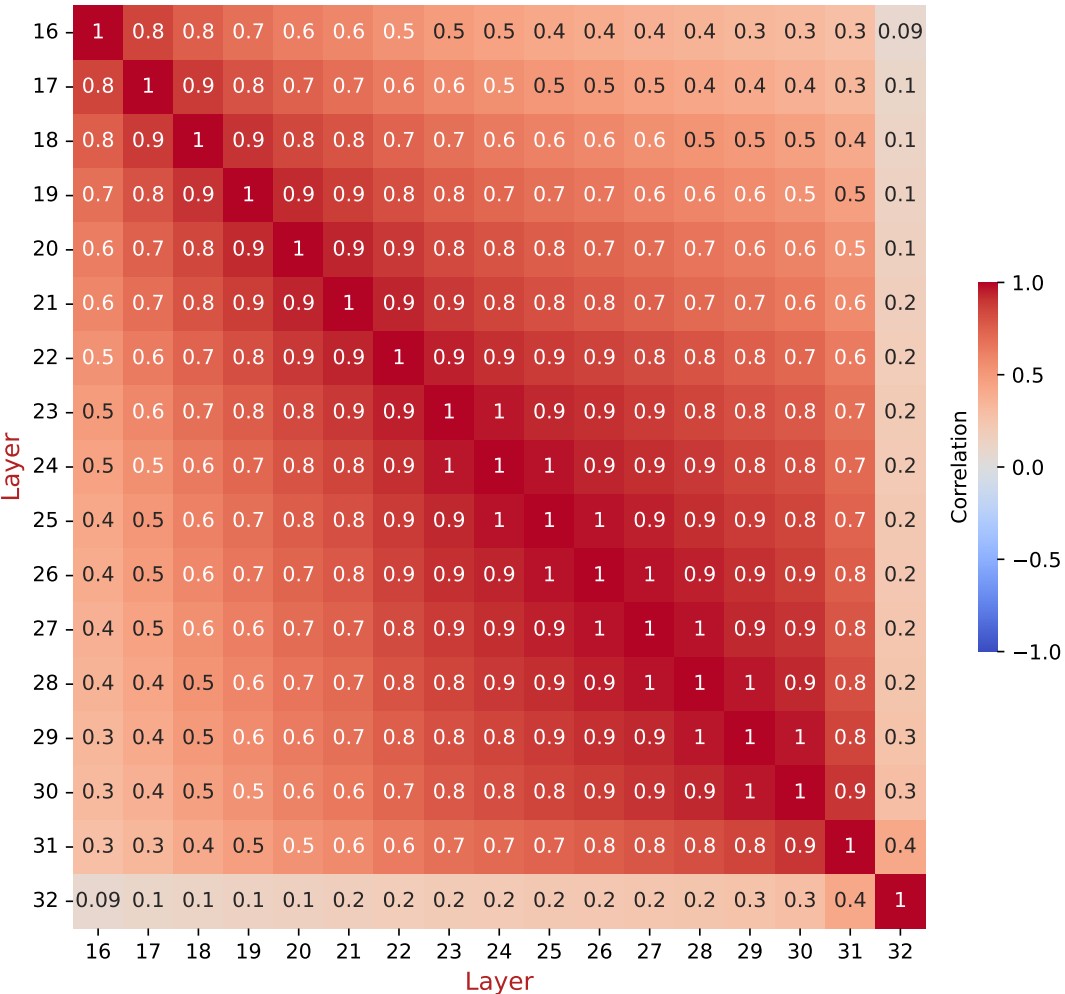

Figure 13: Average Person correlations among the second half of model's hidden layers for Llama3.1

### O.2  Analysis of representation redundancy

In Section 4.3, we hypothesize that highly correlated rows (model's adjacent layers) could cause redundant representations, since they likely encode similar numerical patterns, and thus information.

Here, we present an analysis of representation redundancy, defined as approximate linear dependence among LLM hidden layer embeddings. We computed the Gram matrix $G = HH^T$, where $H$ is the model's residual stream, and analyzed its eigenvalues. Table 17 shows results for the OLMo-2 model (considering the 32nd-to-64th range of hidden layers; Appendix D), averaged on a 100K training input sample. The spectrum reveals a few dominant eigenvalues (around 3-4 modes) followed by many smaller ones, indicating that the embedding space is approximately low-rank. This suggests that, when considering the full matrix ($\mathbb{R}^{33 \times 5120}$ for OLMo-2), most hidden layer representations (rows) are redundant, since only a few rows (or their combinations) contribute meaningful structure. The first mode is by far the most dominant, with a normalized eigenvalue of 0.65, compared to 0.17 for the second. We hypothesize that this leading component might correspond to

next-token prediction representations, while the remaining modes capture secondary structures or auxiliary information. Our hyperdimensional probe aims to capture also these auxiliary latent structures, rather than limiting solely on the single predominant component.

Table 17: Eigenvalues (EV) of the Gram matrix from OLMo-2's residual stream.

| Comp. | EV (mean ± std) | Norm. EV |
|---|---|---|
| 0 | 58084 ± 5293 | 0.650 |
| 1 | 15450 ± 2056 | 0.170 |
| 2 | 5972 ± 608 | 0.070 |
| 3 | 2539 ± 330 | 0.030 |
| 4 | 2057 ± 220 | 0.020 |
| 5 | 1187 ± 166 | 0.010 |
| 6 | 727 ± 119 | 0.010 |
| 7 | 505 ± 83 | 0.010 |
| 8 | 363 ± 59 | 0.000 |
| 9 | 282 ± 48 | 0.000 |
| 10 | 230 ± 37 | 0.000 |
| . . . | . . . | . . . |
| 30 | 30 ± 6 | 0.000 |
| 31 | 27 ± 6 | 0.000 |
| 32 | 22 ± 6 | 0.000 |

### O.3 Silhouette analysis for determining optimal range of clusters

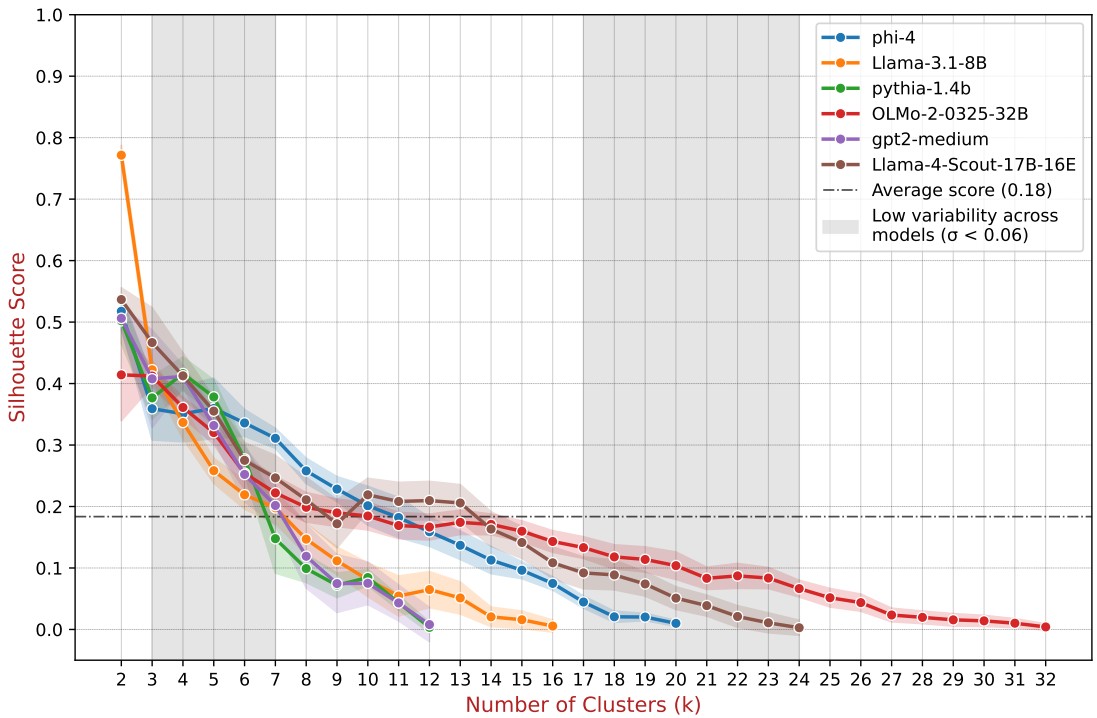

Figure 14: Silhouette scores for varying numbers of clusters, computed using a random sample of 10,000 textual inputs from $\mathcal{S}$. The six language models have varied layer counts (see Table 6), which results in different maximum possible cluster numbers.

## O.4 Distribution of cluster assignments for grouping model's hidden layers

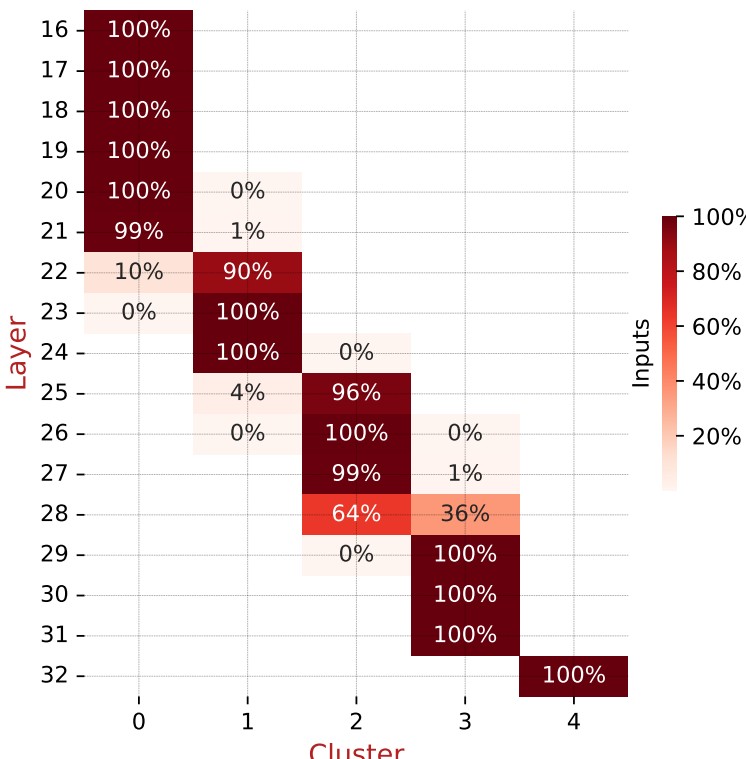

Figure 15: Distribution of model's hidden layers grouped by k-means clustering within the ingestion algorithm $F$ for Llama3.1-8B. It portrays the percentages of cluster assignments across all instances.

## O.5 Ablation study on the dimensionality-reduction steps

This section presents an analysis of skipping the dimensionality reduction steps introduced in Section 4.3. While our VSA-based methodology would work without these compression steps, the overall computational cost of probing would dramatically increase. For example, our ingestion procedure (Appendix B; Section 4.3) reduces the probed OLMo-2's embeddings from $\mathbb{R}^{33 \times 5120}$ to $\mathbb{R}^{5120}$. This allows our neural VSA encoder to have an input dimension $d = 5012$ with only 71M trainable parameters (see Appendix C).

If the two steps are eliminated, and thus the entire residual stream of the model $\mathbb{R}^{33 \times 5120}$ is considered, the encoder receives a flat input vector, creating an input dimension $d = 168960 \in \mathbb{R}^{168960}$. Although the encoder would internally handle feature extraction, since the flattened input holds all the information encoded in the LLM embeddings, this approach would increase the number of trainable parameters to 742 million, representing a tenfold increase. Additionally, adopting a lazy feature extraction stage in an input vector space of size $\approx 10^5$, which is approximately low-rank (see Appendix O.5), would result in a computationally inefficient approach.

Removing one of the two steps, such as sum pooling, should lead to just an increase of the overall computational cost for the encoder ($\mathbb{R}^{5 \times 5120} \mapsto \mathbb{R}^{25600}$; $d = 25600$; 155M trainable parameters; x2), rather than affecting probe's outputs. Further, since our neural VSA encoder is found effective to extract latent features even from our heavily-compressed input representation (Section 5), other dimensionality reduction approaches could also be as effective as ours (Appendix B).

In summary, skipping the compressing steps is possible and the only drawbacks should be the increase of footprint of both the training and inference stages of the VSA-based probing (see also Appendix Q).

## P    Proof of concept for hyperdimensional probe in multimodal settings

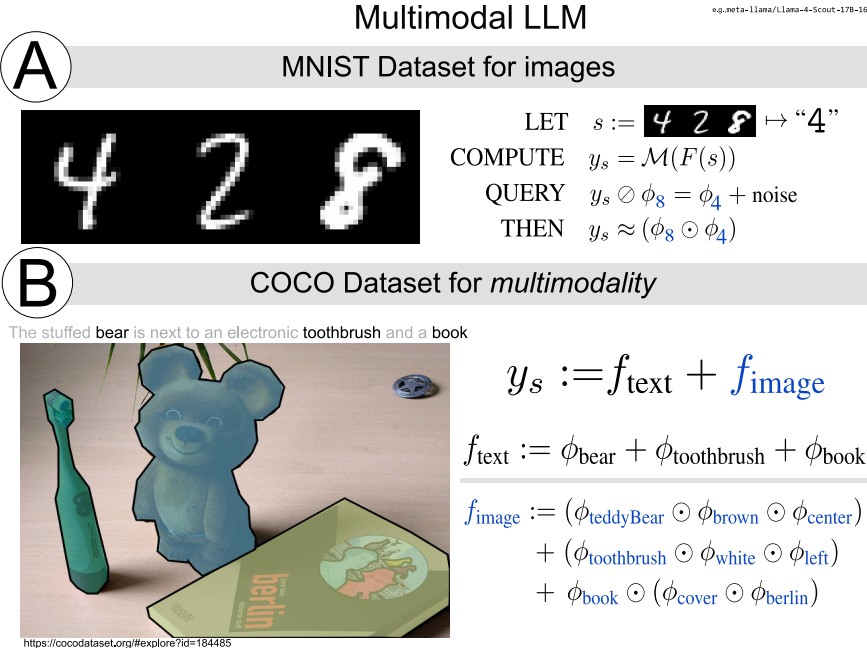

Figure 16: Proof of concept for using hyperdimensional probe in multimodal settings. Figure A shows a complete probing procedure for a MNIST-based mathematical analogy. Figure B exhibits a VSA encodings describing a multimodal input using textual and image features.

## Q    Computational workload

The computational workload of this work is split into two parts: LLM inference (exogenous, Section 4.3) and the training and probing stages of our method (endogenous, Section 4.4 and 4.5).

The exogenous factor, running the Large Language Models, was the most computationally demanding task. For our experiments, we tested six different Large Language Models in inference mode, caching their embeddings for our training phase and probing them dynamically during the inference phase of our work (Figure 1). We worked with LLMs ranging from 355M parameters (GPT-2) to 109B parameters (Llama 4, Scout), using between one and three NVIDIA® A100-80GB GPUs, depending on the model size. Quantization is not employed.

In contrast, the computational demands of our VSA-based methodology is relatively low. The most resource-intensive stage was training our neural VSA encoder, but due to its modest size (ranging from $55M$ to $71M$ parameters, see Appendix C), this process remained lightweight. We performed this training on a single GPU, though it could easily be handled with much less powerful and lower-memory GPUs. Regarding GPU usage, we trained our encoder for approximately 8 hours on each LLM's embeddings, though the process could have been shortened with a less conservative early stopping criterion or by reducing the amount of training data.

The probing stage is then composed of simple vector multiplications (unbinding, Section 3), after loading the heavy LLM and our lightweight trained neural VSA encoder into memory (from 800 MB of the 55M version to 1 GB of the biggest one). Future research could explore even further reducing the latent dimension of our neural VSA encoder (Appendix C) or adopt VSA encodings with lower dimensionality (e.g. $D = 512$, leading to a more lightweight encoder. The time cost of probing is thus comparable to simple LLM inference with a slight increase due to feedforward our lightweight trained model, with the time demand for vector multiplications being negligible. Accordingly, the GPU hours for probing depends mainly on the amount of test data and model size; for example, we used around 95 hours of GPU computation for probing embeddings of Llama 3.1-8B on $\bar{\mathcal{S}} \approx 10^5$, processing each instance in around 3 seconds.

