# OpenReview forum: "Hyperdimensional Probe: Decoding LLM Representations via Vector Symbolic Architectures"
_TMLR — Under review for TMLR_

### Review · Reviewer_eTLR · 2026-03-16

**Summary Of Contributions:**

This paper introduces the Hyperdimensional Probe, a new method for understanding how large language models (LLMs) represent information internally. The method uses Vector Symbolic Architectures (VSAs) to translate LLM embeddings into a structured and queryable space. A small neural network (MLP) is trained to perform this translation, and hypervector operations are then used to extract human-readable concepts. The approach is tested on six different LLMs, from GPT-2 to Llama 4 Scout, using analogy and question-answering tasks. The paper’s main contributions include proposing a new VSA-based probing approach, introducing a layer-agnostic way to process model representations using k-means and pooling, presenting findings about how concepts appear inside LLMs, and suggesting that some QA errors happen because models lose focus on question-related concepts rather than lacking the answer.

The paper has several strengths. It brings together ideas from hyperdimensional computing and LLM interpretability in a novel way, works across multiple models, and includes useful ablation studies. The finding about “concept drift” in question answering is also interesting. However, there are some limitations. The method relies on a predefined concept vocabulary, which limits its use in open-ended settings. There is also no causal validation to show that the identified concepts directly influence model behavior. In addition, the QA encoder’s performance is moderate (cosine similarity 0.44 and accuracy 0.70), and the comparisons with other interpretability methods, especially sparse autoencoders, are limited.

**Audience:**

Yes

**Audience Explanation:**

LLM interpretability is a highly active area, and bridging VSA/hyperdimensional computing with neural probing is a genuinely novel angle. The finding that a model can internally encode the correct answer yet fail to output it (Llama 4 Scout) is practically relevant. The concept drift finding in QA, if further validated, could be valuable to the alignment/ELK community.

**Broader Impact Concerns:**

No major ethical concerns are identified. The paper is a basic research contribution to LLM interpretability and does not present clear harmful applications. The authors also state that the code and dataset will be publicly released, which supports transparency and reproducibility.

A formal Broader Impact Statement may not be necessary, but the paper could briefly note that interpretability tools might also be used to better understand and potentially influence model behavior.

**Claims And Evidence:**

Yes

**Claims Explanation:**

The analogy experiments are well designed, using 114K examples across 52 domains and six different LLMs, and they include proper null and unrelated baselines. The reported improvement—about 83% probing@1 compared to about 54% for DLA—appears credible and well supported by the results.

The QA analysis (Section 6) is weaker. The encoder accuracy of 0.70 is relatively modest, and the “concept drift” result, while interesting, is based on only one model (Llama 3.1) and a fairly weak Spearman correlation (ρ ≈ 0.2). Still, the main claims about the analogy experiments are well supported.

**Requested Changes:**

The QA encoder in Section 6 performs much worse than the analogy encoder, with only 0.44 cosine similarity and 0.70 binary accuracy compared to about 0.88 and 0.93 for the analogy setting. The authors should either improve the QA encoder or clearly limit the strength of the conclusions drawn from this section. In particular, the “concept drift” result should not be presented as strong evidence unless the encoding quality is higher.

The paper also compares the method only with DLA (Logit Lens) and does not include comparisons with sparse autoencoders (SAEs), even though SAEs are discussed as a key related method. The explanation in Appendix G—that SAE comparisons require manual feature labeling—is reasonable but not sufficient. The paper should include at least a partial or qualitative comparison, or more clearly discuss the limitations of not evaluating against SAEs.

Finally, the method depends on a predefined concept vocabulary, which limits its use in open-ended settings. The paper should explain more clearly how such a concept codebook could be built for open-domain tasks and what happens when important concepts are not included in the vocabulary.

---

### Review · Reviewer_VyLY · 2026-06-30

**Summary Of Contributions:**

This paper argues that understanding LLM vector spaces requires integrating both input-oriented feature extraction and output-distribution inspection. To this end, the authors introduce the Hyperdimensional Probe, a hybrid supervised probe that combines symbolic representations with neural probing. Experiments show that the proposed method consistently extracts meaningful concepts across LLMs, embedding sizes, and experimental setups, uncovering concept-driven patterns in analogy-oriented inference and QA-focused text generation.

**Strengths:**
- This paper is well-organized and interesting in bridging input-oriented feature extraction with output-distribution inspection.
- The use of Vector Symbolic Architectures and hypervector algebra provides a structured proxy space, offering an interesting and novel alternative to existing supervised probes.
- The authors conduct experiments on multiple LLMs with different sizes and architectures, and consider both analogy-style completion and QA-oriented text generation.

**Weaknesses:**
- The claim about “concept-driven patterns” seems somewhat overstated. The current evidence primarily shows that the probe can recover structured key- and target-aligned concepts in predefined, template-based analogy tasks. It does not yet demonstrate broader concept-driven inference, such as compositional concept use, in LLMs.
- The proposed method uses a high-capacity supervised encoder: a three-layer MLP with 55M–71M parameters trained to map LLM embeddings to VSA targets. Therefore, the current results do not clearly distinguish between decoding concepts already present in the LLM representation and learning task-specific regularities from the training data.
- The VSA method uses a predefined concept set aligned with the analogy task, while SAEs learn features in a bottom-up. Thus, comparing the two methods mainly on target-concept retrieval may not be a fair evaluation.
- The work lacks causal validation. The results show that concepts are decodable from LLM representations, but they do not show that these concepts are used. Interventions such as activation patching, concept steering should be applied to the extracted concepts to test whether they causally affect model behavior.

Minor: some typos, please check them.

**Additional Comments:**

NA

**Audience:**

Yes

**Audience Explanation:**

Some TMLR readers interested in LLM interpretability, probing methods, and representation analysis would find the proposed VSA-based probing framework and its empirical findings useful.

**Claims And Evidence:**

No

**Claims Explanation:**

The experiments show that the proposed Hyperdimensional Probe can extract target-aligned concepts across several LLMs and settings. However, because the method relies on a supervised VSA encoder, it remains unclear whether the probe is decoding concepts already present in the LLM representations or learning task-specific structure from the training data. Second, the paper overclaims concept-level generalization without stronger held-out experiments or causal validation, such as steering or activation patching.

**Requested Changes:**

Please see my comments under Weaknesses. I would encourage the authors to make the claims more precise and to provide stronger causal validation.

---

> ### Author Response · Authors · 2026-07-20
> **Authors' comments**
>
> We appreciate your thoughtful comments. We are pleased that you found our research efforts to bridge input-oriented feature extraction with output-distribution inspection using Vector Symbolic Architectures (VSAs) to be both interesting and novel, recognizing our contribution in providing an alternative to existing supervised probes.
>
> Here, we would like to clarify some aspects:
>
> ---
>
> ## 1. Claim on Concept-Driven Patterns
>
> > **Reviewer Comment:** *"The claim about concept-driven patterns seems somewhat overstated."*
>
> The revised version relaxes the claims concerning discovered concept-driven patterns (see the blue text in the manuscript). By experimenting with two distinct experimental setups and diverse language models, our work uncovers non-trivial insights into the inference of language models:
>
> * **Controlled, template-based analogy tasks (Section 5):** We unveil critical differences in concept-driven compositionality among diverse language models and input types. For instance, Section 5.2 shows that OLMo-2 possesses a more contextually rich latent structure (i.e., compositionality) compared to GPT-2 (**Table 1**). This section, which now integrates additional material from the former Appendix F.2, also identifies more frequent blank representations tied to specific input domains, which may depend on the specific type of reasoning required.
> * **Question-Answering inference (Section 6):** We track concept-related dynamics immediately before and after the model's text generation in a QA-driven inference of two diverse language models (i.e., Llama-3.1 and GPT-2). This reveals non-trivial insights into the model's decoding process, such as the proportion of question- and answer-related concepts during LLM inference. Based on these uncovered patterns, we hypothesize that Llama's wrong answers might stem from losing focus on question-related concepts rather than answer-related ones. Additionally, this analysis yields model-wise insights by highlighting the contrasting workings of GPT-2.
>
> While demonstrating the operational usage of semantic information from neural representations remains an open challenge in LLM interpretability, doing so falls outside the scope of our work. Beyond target concept retrieval (**Figure 4**), our primary evaluation focus is compositional concept extraction (**Tables 1, 3, and 4**), specifically, uncovering how semantic concepts are compositionally encoded within the LLM's vector space.
>
> To this end, our experiments provide concrete evidence of inference-driven compositionality by analyzing diverse sets of semantic concepts directly within these latent spaces, both in a next-token prediction setup (Section 5) and during active text generation (Section 6).
>
> ---
>
> ## 2. Concept-Level Generalization
>
> > **Reviewer Comment:** *"the paper overclaims concept-level generalization without stronger held-out experiments."*
>
> All of our experimental findings are derived from an out-of-distribution evaluation framework, serving as a rigorous held-out test. Specifically, we test the trained encoders on textual inputs featuring syntactic structures that were completely unseen during training (see **Figure 2**). Consequently, they can be seen as held-out tests with respect to the distribution of LLM embeddings learned by our trained encoder.
>
> On the other hand, as the encoder maps semantic concepts with their corresponding neural representations, it must be exposed to a few examples of the latent features being targeted, whether drawn from in-domain or out-of-domain distributions (Appendix I.3). In other words, it cannot extract semantic concepts that it has never encountered.
>
> ---
>
> ## 3. Lack of Causal Validation
>
> > **Reviewer Comment:** *"The work lacks causal validation."*
>
> In its current implementation, our approach relies on a unidirectional transformation from neural embeddings to VSA encodings, which inherently precludes causal validation. While intervention-based analysis is beyond the primary scope of our work, we explicitly acknowledge this constraint in our limitations and introduction sections.
>
> As a natural evolution, we have conducted experiments utilizing a bidirectional transformation to enable VSA-based steering (now reported in Appendix I.4). This shifts the VSA-based framework from an encoder-only paradigm to an encoder-decoder architecture:
>
> $$\mathcal{M}: \mathbb{R}^d \to \\{-1,1\\}^D \to \mathbb{R}^d$$

---

> ### Author Response · Authors · 2026-07-20
> **Authors' comments (...)**
>
> We have already obtained promising preliminary results in steering autoregressive text generation. To manipulate the language of text generation, our VSA-based approach learns a reversible mapping between a given language and its specific directions or subspaces within the LLM space. We train an autoencoder-like model with a combined loss function designed to reconstruct the LLM embeddings while internally representing the target VSA concept:
>
> $$e(\text{"Die Hauptstadt Frankreichs ist Paris"}) \mapsto \phi_{\text{german}} \mapsto e(\text{"Die Haup..."})$$
>
> This training stage achieved an average test-set binary accuracy of 0.99 for VSA concepts during the encoder stage, and a cosine similarity of 0.96 when reconstructing the original LLM embeddings in the decoder stage.
>
> Afterwards, to steer the text generation, we project the VSA concepts back into the LLM representation space using the decoder stage (yielding, for example, $e_{\text{german}}$ and $e_{\text{spanish}}$). We then subtract the source language concept from the hidden representations while adding the target language feature.
>
> For example, taking the last-token embeddings of the French sentence *"Est situé dans l'état de l'Utah"*, we can successfully steer the model's text generation toward Spanish, producing: *"El Chorro es un pueblo de la comarca de la Costa del Sol"*. The table below presents a language accuracy heatmap highlighting these preliminary cross-lingual steering capabilities.
>
> ### Language Accuracy Heatmap under VSA-based Steering
>
> |  | **DE** | **EN** | **ES** | **FR** | **IT** | **JA** | **RU** | **ZH** |
> | --- | --- | --- | --- | --- | --- | --- | --- | --- |
> | **DE** | 1.00 | 0.61 | 0.39 | 0.40 | 0.70 | 0.38 | 0.47 | 0.70 |
> | **EN** | 0.88 | 1.00 | 0.64 | 0.75 | 0.85 | 0.69 | 0.74 | 0.80 |
> | **ES** | 0.85 | 0.84 | 1.00 | 0.84 | 0.88 | 0.49 | 0.74 | 0.78 |
> | **FR** | 0.77 | 0.63 | 0.50 | 1.00 | 0.80 | 0.45 | 0.56 | 0.67 |
> | **IT** | 0.38 | 0.71 | 0.20 | 0.33 | 1.00 | 0.24 | 0.37 | 0.56 |
> | **JA** | 0.96 | 0.96 | 0.84 | 0.93 | 0.93 | 1.00 | 0.88 | 0.85 |
> | **RU** | 0.80 | 0.63 | 0.65 | 0.73 | 0.75 | 0.36 | 1.00 | 0.82 |
> | **ZH** | 0.59 | 0.58 | 0.21 | 0.50 | 0.69 | 0.27 | 0.34 | 1.00 |
>
> Fully incorporating a steering-based framework grounded in VSAs within the main text would require detailing extensive methodological shifts (e.g., formalizing the encoder-decoder architecture, the mathematical nuances of the steering procedure, and dedicated evaluation metrics), drastically altering the paper's scope.
>
> In summary:
>
> 1. The current VSA-based methodology serves as a foundational building block (unidirectional transformation) toward actively steering LLM generation via algebraic symbolic concepts (bidirectional).
> 2. Fully integrating a generative steering use case into the main body would compromise the focus and readability of the paper.
> 3. We are actively developing a standalone VSA-based steering methodology for a future dedicated publication, and have added steering results to the appendix of the current manuscript.
>
> ---
>
> ## 4. Probe-Induced Learning
>
> > **Reviewer Comment:** *"current results do not clearly distinguish between decoding concepts already present in the LLM representation and learning task-specific regularities from the training data"*
>
> While distinguishing probe-induced learning from actual information decoding remains a persistent challenge in probe-based interpretability analysis (e.g., Belinkov, 2022), our work adopts several mitigation strategies to reduce the confounding effects of learning regularities from training data:
>
> * **Testing on diverse textual inputs (Figure 2; Section 5.1):** Our methodology is experimentally validated using textual inputs with syntactic structures that differ from those seen during the training stage:
>     - $\mathcal{\bar{S}} \neq \mathcal{S}$ for the analogy setup (Section 5.1), and
>     - $\mathcal{\bar{Q}} \neq \mathcal{Q}$ for the QA-based setup (Section 6).
>
> Thus, our approach performs information decoding (i.e., probing) from the vector representation of completely different textual inputs. This allows us both to control for confounding effects and to validate the probe’s out-of-domain performance (Appendix I.3). While sharing the same underlying semantic concepts, they consist of distinct texts with a different syntactic structure. Consequently, they can be seen as held-out tests with respect to the distribution of LLM embeddings learned by our trained encoder.

---

> ### Author Response · Authors · 2026-07-20
> **Authors' comments (...)**
>
> * **Diverse experimental settings:** We report experimental findings in two settings with fundamentally different probing objectives and methodological procedures to construct the VSA-based proxy space. While analogy-based tasks (Section 5) provide a controlled testbed using inputs with known templates, Section 6 employs the same pipeline but constructs the proxy space using an automatic procedure grounded in lexical semantics. This complementary evaluation proves that our methodology is robust and not tied to any single proxy-space construction.
> * **Indirect learning objective (Section 2):** Our approach learns a transformation from LLM embeddings to a controlled proxy space (i.e., a composition of concepts, structured as VSA encodings) rather than directly targeting task-specific features like standard supervised probes.
> * **Control tests (Appendix F.1):** We conduct two validation experiments to assess whether the proposed approach decodes information from LLM embeddings rather than producing meaningful outputs under arbitrary conditions.
>     1. We evaluate a **null model** by providing the trained probe with randomly permuted LLM embeddings (`Permuted baseline` in Table 8).
>     2. We attempt to extract concepts that are unrelated to the probed textual inputs from the generated proxy space (`Unrelated baseline` in Table 8).
>
> Both experiments yield meaningless outputs, providing evidence that the probe relies on information encoded in the original LLM embeddings and does not recover arbitrary concepts. While these sanity checks cannot conclusively rule out probe-induced learning, a fundamental limitation of probing methods, they support the interpretation that our results primarily reflect information decoded from the embedding space rather than artifacts of the probe itself.
>
> However, it is important to recognize a fundamental limitation of all probing approaches: because the human-interpretable information encoded in LLM embeddings is not explicitly known, no probing method can conclusively rule out the influence of probe-induced learning.
>
> ---
>
> ## 5. Comparison with SAEs
>
> > **Reviewer Comment:** *"comparing the two methods [SAEs, VSA] mainly on target-concept retrieval may not be a fair evaluation."*
>
> The experimental comparison with Sparse Autoencoders (SAEs) was included in response to Reviewer eTLR's request. Although Sections 2 and 5.4 discuss the fundamental methodological differences between SAEs and our VSA-based approach, most notably the bottom-up, unsupervised discovery of latent features by SAEs, we designed the experimental protocol to provide a comparison that is as fair as possible while respecting the experimental setup underlying our work and the typical operating regime of SAE-based analyses.
>
> On the other hand, Section 5.4 does present a comprehensive comparison between the two methods across the two core aspects investigated in our work: target concept retrieval (**Figure 4**) and concept compositionality (**Table 4**).
>
> This comparison aims to show that SAEs inherently learn noisier, less structured latent representations when restricted to a constrained conceptual feature space. Testing alternative SAE setups (such as substituting DLA-based feature extraction with input clustering) does not affect this central thesis because their fundamental behavior remains unchanged across different configurations. Consequently, our findings suggest that SAE-based interpretability is less suited for bounded conceptual frameworks, highlighting the utility of our VSA-based approach in structured settings.

---

### Review · Reviewer_ToBn · 2026-07-09

**Summary Of Contributions:**

This paper proposes Hyperdimensional Probe, an interpretability framework for decoding LLM internal representations using Vector Symbolic Architectures (VSAs). The method first compresses hidden-layer residual-stream embeddings, then trains a neural encoder to map these embeddings into a bounded VSA proxy space, where concepts can be queried through hypervector algebra. Experiments on analogy-completion tasks and SQuAD-style QA show that the probe can recover meaningful input and output-related concepts from different LLMs, often revealing target concepts even when next-token predictions or logit-lens analysis fail.Abundant experiments are implemented to demonstrate its performance.

**Additional Comments:**

Questions:

- 1 How can the authors better distinguish true information decoding from probe-induced learning, especially given the synthetic analogy corpus and predefined VSA codebook?
- 2 Does the extracted VSA concept have any causal effect on the final prediction, or is it only correlated with the hidden representation?

**Audience:**

Yes

**Audience Explanation:**

This paper focuses on a very interesting and timely topic that I believe will be of interest to many readers.

**Claims And Evidence:**

Yes

**Claims Explanation:**

Strengths:

- 1 The paper proposes an interesting way to combine Vector Symbolic Architectures with neural probing for LLM explanation.
- 2 The paper is generally well-written.

**Requested Changes:**

Weakness:

- 1 The method can decode concepts from hidden representations, but it does not show that these decoded concepts causally affect the model’s outputs. Since the probe is a one-way mapping from LLM embeddings to VSA encodings, the analysis remains mostly correlational rather than intervention-based.
- 2 The construction of the VSA proxy space may introduce probe-induced learning and information leakage. Although the authors test on a different textual template, the neural VSA encoder is still trained on a synthetic analogy corpus built from the same predefined concept codebook and similar key–value structures. Since the train/test split is random over generated analogy instances, the probe may partially learn regularities of the synthetic VSA labels or memorized key–value associations, instead of purely decoding information already present in the LLM representations. A stronger validation would require held-out concepts/domains or relation types, and a comparison showing that the probe still works when the symbolic codebook is not closely aligned with the training corpus.
- 3 Most evidence comes from synthetic analogy-completion tasks and one QA setting. The comparison is mainly against DLA/logit lens, while stronger comparisons with SAEs or other supervised probing methods are not fully developed.

---

> ### Author Response · Authors · 2026-07-20
> **Authors' comments**
>
> We appreciate your recognition that our work proposes an interesting way to combine neural probing with Vector Symbolic Architectures (VSAs) for LLM explanation. In response to your comments, we would like to offer the following clarifications:
>
> ---
>
> ## 1. Contribution and Scope: Correlation vs. Intervention
>
> > **Reviewer Comment:** *"the analysis remains mostly correlational rather than intervention-based"*
>
> The revised manuscript more clearly discusses the contribution, scope, and limitations of our investigation, framing our experimental findings as observational evidence regarding the semantic information encoded in LLM latent representations rather than as causal claims.
>
> As you correctly noted, the current implementation of our probe consists of a one-way mapping from LLM embeddings to VSA encodings, which serve as a proxy for analyzing the LLM representation space. Consequently, this design does not support intervention-based analyses, such as causal validation, a limitation that is explicitly acknowledged in the manuscript (**Section 1** and **Section 7**).
>
> While intervention-based analysis is beyond the primary scope of our work, we have conducted experiments utilizing a bidirectional transformation to enable VSA-based steering (now reported in **Appendix I.4**). This shifts the VSA-based framework from an encoder-only paradigm (**Section 4.4**) to an encoder-decoder architecture:
>
> $$\mathcal{M}: \mathbb{R}^d \to \\{-1,1\\}^D \to \mathbb{R}^d$$
>
> In the revised version, we describe promising results obtained in steering autoregressive text generation. To manipulate the language of text generation, our VSA-based approach needs to learn a reversible mapping between a given language and its specific directions or subspaces within the LLM space. We train an autoencoder-like model with a combined loss function designed to reconstruct the LLM embeddings while internally representing the target VSA concept:
>
> $$e(\text{“Die Hauptstadt Frankreichs ist Paris”}) \mapsto \phi_{\text{german}} \mapsto e(\text{“Die Haup...”})$$
>
> This training stage achieves an average test-set binary accuracy of **0.99** for VSA concepts during the encoder stage, and a cosine similarity of **0.96** when reconstructing the original LLM embeddings in the decoder stage.
>
> Afterwards, to steer the text generation, we project the VSA concepts back into the LLM representation space using the decoder stage (yielding, for example, $e_{\text{german}}$ and $e_{\text{spanish}}$). We then subtract the source language concept from the hidden representations while adding the target language feature.
>
> For example, taking the last-token embeddings of the French sentence *"Est situé dans l'état de l'Utah"*, we can successfully steer the model's text generation toward Spanish, producing: *"El Chorro es un pueblo de la comarca de la Costa del Sol"*. **Table 1** presents a language accuracy heatmap highlighting these preliminary cross-lingual steering capabilities.
>
> ### Table 1: Language Accuracy Heatmap under VSA-based Steering
>
> |  | **DE** | **EN** | **ES** | **FR** | **IT** | **JA** | **RU** | **ZH** |
> | --- | --- | --- | --- | --- | --- | --- | --- | --- |
> | **DE** | 1.00 | 0.61 | 0.39 | 0.40 | 0.70 | 0.38 | 0.47 | 0.70 |
> | **EN** | 0.88 | 1.00 | 0.64 | 0.75 | 0.85 | 0.69 | 0.74 | 0.80 |
> | **ES** | 0.85 | 0.84 | 1.00 | 0.84 | 0.88 | 0.49 | 0.74 | 0.78 |
> | **FR** | 0.77 | 0.63 | 0.50 | 1.00 | 0.80 | 0.45 | 0.56 | 0.67 |
> | **IT** | 0.38 | 0.71 | 0.20 | 0.33 | 1.00 | 0.24 | 0.37 | 0.56 |
> | **JA** | 0.96 | 0.96 | 0.84 | 0.93 | 0.93 | 1.00 | 0.88 | 0.85 |
> | **RU** | 0.80 | 0.63 | 0.65 | 0.73 | 0.75 | 0.36 | 1.00 | 0.82 |
> | **ZH** | 0.59 | 0.58 | 0.21 | 0.50 | 0.69 | 0.27 | 0.34 | 1.00 |
>
> In summary, while the revised manuscript more clearly defines the contribution, scope, and limitations of our experimental analysis, **Appendix I.4** demonstrates the practical applicability of the VSA-based methodology for steering LLM text generation.
>
> ---
>
> ## 2. Probe-Induced Learning and Information Leakage
>
> > **Reviewer Comment:** *"the probe may partially learn regularities of the synthetic VSA labels or memorized key–value associations."*
>
> Our work addresses the issue of probe-induced learning through several mitigating strategies designed to exclude memorized key–value associations. In addition to evaluating the probe on structurally diverse inputs, we further demonstrate the generality of our VSA-based methodology in **Section 6**. In this setting, the VSA proxy space is constructed using an entirely different procedure to address a substantially different probing objective.

---

> ### Author Response · Authors · 2026-07-20
> **Authors' comments (...)**
>
> As shown in **Section 6 (Equation 5)**, the training process in the QA-based setting progressively associates textual inputs with their corresponding lexical feature representations automatically, rather than relying on predefined associations:
>
> $$ \text{``What was the name"} \mapsto (\phi_{\text{name}}) $$
>
> $$  \text{``What was the name of the ship"} \mapsto (\phi_{\text{name}} + \phi_{\text{ship}}) $$
>
> $$ \text{``What was the name of the ship that Napoleon"} \mapsto (\phi_{\text{name}} + \phi_{\text{ship}} + \phi_{\text{napoleon}}) $$
>
> $$ \dots $$
>
> This results in a fundamentally different strategy for constructing the VSA-based proxy space, moving completely beyond the predefined key–value associations used in the analogy-based setting, such as those illustrated in **Section 4 (Equation 2)**:
>
> $$\text{``Denmark : krone = Mexico : peso "} \mapsto (\phi_{\text{denmark}} \odot \phi_{\text{krone}}) + (\phi_{\text{mexico}} \odot \phi_{\text{peso}})$$
>
> Specifically, the neural VSA encoder for QA-style inputs learns a VSA-based proxy space (i.e., an ensemble of semantic concepts represented as VSA-based encodings) containing a variable number of concepts (up to eight) using only the bundling operation. This relies purely on lexical semantics without any predetermined associations, resulting in dynamic, variable, and template-free output representations.
>
> This allow us to validate our methodology on a more realistic setting compared to the controlled testbed (**Sections 4–5**), which relies on inputs with predefined semantic templates. Afterward, the probing objective of the trained encoder—where textual inputs also include the SQuAD context (see **Equation 8 in Appendix J**)—is to retrieve as many relevant concepts as possible and benchmark them against those associated with the question and answer during text generation (**RQ6**). This experimental setup extends the earlier analogy-style experiments (**Section 5**), where the encoder learned a proxy space by relying on a fixed structure formatted with known key-value associations to answer different research questions (**RQ1–RQ5**).
>
> #### Information Leakage
> On the other hand, we also conducted an experimental analysis on held-out extraction to control for memorized associations in the analogy-based setup, as further described in the extended limitation section (**Appendix A**):
>
> 1.  **Tables 1, 3, and 10** report on the actual concepts identified by our probing method. The label `Key values` denotes instances where the probe retrieves a key of an analogy pair alongside a concept linked to it in a different domain (see *Australia* in Section 4.1). This outcome can be viewed as an artifact of our probe, revealing the confounding influence of memorized key-value associations. Nevertheless, such cases constitute only a small fraction—**2% of the 114,099 textual inputs** processed across all models, covering *Key | Key Values*, *Example Value | Key Values*, and *Key Values | Target* shown in Table 1.
>
> 2. We introduce a control test on information decoding by probing the VSA-based proxy space (i.e., applying the unbinding operation on the generated VSA encodings) with concepts unrelated to the inputs. This is labeled as the `unrelated baseline` in **Appendix F.1** and includes key-value associations from completely different domains. This experiment indicates that applying VSA-based probing using concepts irrelevant to the input texts results in meaningless outputs, providing further indirect validation of genuine information decoding rather than probe-induced learning.
>
> ---
>
> ## 3. Responses to Specific Queries
>
> ### Q1: How can the authors better distinguish true information decoding from probe-induced learning?
>
> Distinguishing probe-induced learning from genuine information decoding remains an ongoing challenge in probe-based interpretability (e.g., Belinkov, 2022). Since the human-interpretable information encoded within LLM embeddings is not explicitly known, no probing method can completely rule out probe-induced learning. Although we acknowledge this inherent limitation in **Section 7** and **Appendix F.1**, our work employs several mitigation strategies to minimize its confounding effects:
>
> * **Diverse experimental settings:** Although the analogy-based tasks (**Section 5**) provide a controlled testbed for validating our framework, their VSA proxy space is constructed from template-driven structures. To address concerns that the probe might simply be learning these predefined structures or memorizing key-value associations, we evaluate our method in a fundamentally different setting. Our QA-based experiments (**Section 6**) employ the same VSA probing pipeline but construct and probe the proxy space using an entirely different procedure. This complementary evaluation proves that our methodology is robust and not tied to any single proxy-space construction or template-based representation.

---

> > ### Author Response · Authors · 2026-07-20
> > **Authors' comments (...)**
> >
> > * **Testing on diverse textual inputs (Figure 2; Section 5.1):** Our trained encoders are tested using textual inputs with syntactic structures that differ from those seen during their training stage:
> >
> >      - $\bar{\mathcal{S}} \neq \mathcal{S}$ for the analogy setup (Section 5.1)
> >
> >      - $\bar{\mathcal{Q}} \neq \mathcal{Q}$ for the QA-based setup (Section 6)
> >
> >     Thus, we probe information from the vector representations of completely different textual inputs. This dual design controls for confounding effects while validating the probe's out-of-distribution performance (**Appendix I.3**). Because these inputs share semantic concepts but differ in syntactic structure, they act as true held-out tests for the LLM embedding distribution learned by our trained encoder.
> > * **Indirect learning objective (Section 2):** Our approach learns a transformation from LLM embeddings to a controlled proxy space (i.e., a composition of VSA-based concepts). This mimics the dictionary learning paradigm of Sparse Autoencoders, avoiding directly targeting task-specific features like standard supervised probes.
> > * **Control tests (Appendix F.1):** To verify that our approach decodes genuine information from the LLM embeddings rather than generating arbitrary outputs, we perform two validation experiments. First, we test a null model by feeding the trained probe randomly permuted embeddings (`Permuted baseline`, Table 8). Second, we attempt to extract concepts completely unrelated to the input text from the generated proxy space (`Unrelated baseline`, Table 8). Both baselines yield meaningless results, confirming that the probe relies on the structure of the original embeddings. While these sanity checks cannot entirely rule out probe-induced learning, they strongly suggest our results reflect decoded representations rather than artifacts of the probe (see **Appendix A** for an analysis of the probe's artifacts).
> >
> > ### Q2: Does the extracted VSA concept have any causal effect on the final prediction, or is it only correlated with the hidden representation?
> >
> > In the revised version, we clarify that our experimental findings provide observational evidence of the semantic content in LLM representations, rather than proving a causal, mechanistic link between these extracted concepts and the model's outputs.
> >
> > While our primary objective is to identify which semantic concepts are encoded in the LLM's latent space and correlate them with downstream behavior (such as incorrect outputs), **Appendix I.4** introduces an intervention-based analysis using Vector Symbolic Architectures (VSAs) to steer text generation. These encouraging results demonstrate the causal impact of a bidirectional transformation grounded in VSA representations, showcasing the viability of a VSA-based methodology for actively controlling LLM generation.
> >
> > ---
> >
> > ## 4. More Extensive Experimental Comparisons
> >
> > > **Reviewer Comment:** *"stronger comparisons with SAEs or other supervised probing methods are not fully developed."*
> >
> > **Sections 5.3 and 5.4** present experimental comparisons of our VSA-based approach with DLA and SAEs across our two main evaluation dimensions: target concept retrieval (**Figure 4**) and concept compositionality (**Tables 1, 3, and 4**).
> >
> > In addition to describing how our setup was adapted to support DLA- and SAE-based analyses, these sections evaluate how both methodologies behave under the same experimental setting. In bounded frameworks with constrained feature spaces, our approach yields more structured and less noisy findings than SAE-based analyses. This discrepancy stems from their fundamental operating principles, particularly the bottom-up, unsupervised nature of SAE feature discovery.
> >
> > To ensure an equitable evaluation, the protocol in **Section 5.4** balances the assumptions of our setup with the standard operating conditions of SAEs. Investigating alternative SAE configurations, such as substituting DLA-based feature extraction with input clustering, would not change our central research question. Because our comparison evaluates SAEs within a constrained feature space, their bottom-up learning paradigm remains unchanged regardless of alternative configurations.
> >
> > On the other hand, we position our method as a **hybrid supervised probe**, as it combines the supervised learning paradigm of traditional probes with the sparse, dictionary-based representation principle of SAEs. This design intentionally bridges two traditionally separate fields: supervised probing and sparse representation learning. The contribution therefore lies in the integration of VSA-based symbolic representations with supervised probing, rather than in proposing an alternative variant of conventional probes or SAEs. Consequently, direct comparisons with non-VSA probing methods would address a different methodological question, as such methods neither rely on VSA representations nor operate within the same concept-based proxy-space framework developed in this work.

---

### Author Response · Authors · 2026-04-28
**New revised version and authors' comments**

We sincerely appreciate the thoughtful and constructive feedback from reviewer eTLR. We have now *uploaded a revised version* and have *improved (green-highlighted texts) the following aspects*:

# (1) Comparison with Sparse Autoencoders🧑🏻‍🔬
The revised manuscript **now includes an *experimental comparison* between our VSA-based results and those derived from *Sparse Autoencoders*** (SAEs). This new analysis, described in **Section 5.4** (and in Figure 4 and Table 3) covers the *full SAE workflow*, from *training autoencoders* on neural embeddings for each of the six language models to *feature naming* and *concept extraction*.

- The SAE-based comparison shows that the *lack of relevance-based feature filtering*, arising from SAEs’ inherent objective of uncovering latent structure in an unsupervised manner, can lead to **noisier interpretability analyses in bounded conceptual frameworks**, such as those required by our research questions (Section 1).
-  For example, given the textual input *“king is to queen as son is to,”* SAE-based analysis on OLMo-2 identifies concepts outside the bounded in-context set, extracting concepts such as man, human, girl, boy, and gender. While these are conceptually related to the input, this unconstrained output **prevents direct answers to our research questions** regarding the presence of specific in-context concepts in a given input (RQ1–RQ3).
- This further experimental comparison indicates that VSA probing not only yields deeper insights than conventional logit-based analysis (Section 5.3) but is also particularly **well-suited to experimental settings requiring a constrained or structured conceptual feature space**, addressing a key limitation of SAEs’ unbounded feature space.

See Appendix G for further discussion of our experimental comparison decisions, including the absence of additional steps required for Direct Logit Attribution (DLA) and the implications of SAE’s bottom-up paradigm.


# (2) Performance of the encoder for the QA setting 📉
The encoders employed in the two experimental settings *are not directly comparable*, as they **pursue different probing objectives and adopt distinct strategies for constructing the targeted outputs**, the VSA-based representations.
1. The **neural VSA encoder for analogy-style inputs** (Section 5) merges exactly four concepts through a fixed encoding procedure consisting of two binding operations ($\odot$) that form key–value pairs, followed by a final bundling operation ($+$).
    - This process relies on a *fixed and predefined structure for the output representation*.
    - The probing objective of analogy encoder is to determine which of the four encoded concepts are presented during inference (RQ1-RQ3).
2. In contrast, the **neural VSA encoder for QA-style inputs** (Section 6) generates VSA representations containing a variable number of concepts, up to eight, using only the bundling operation, also known as superposition.
    - This results in *dynamic and variable output representations*.
    - Its probing objective is to retrieve as many relevant concepts as possible and benchmark them against those associated with the question and answer during text generation (RQ6).

Consequently, the *lower performance of the QA encoder is expected*, as it must handle:
- *more dynamic* and *noisier output encodings*, where multiple concepts are merged via superposition; and
- a *variable number of concepts*, in contrast to the fixed structure of the analogy encoder.

The QA-based setting introduces a potential bundling-induced information loss. Merging multiple concepts into a single VSA representation generates interference noise that increases with the number of bundled items (Section 3).
As more concepts are superposed, interference increases and the signal-to-noise ratio decreases, leading to less separable representations. Consequently, the encoder learns a more diffuse mapping from neural embeddings to VSA encodings.

Nevertheless, *hypervectors in VSAs are designed to be noise-tolerant*, so approximate recovery of underlying concepts remains possible even under moderate degradation in accuracy. Reducing or fixing the number of bundled concepts may improve encoder training stability and reconstruction quality. However, such constraints could limit representational flexibility and reduce the system’s ability to learn disentangled subspace structures useful for downstream probing tasks.

---

> ### Author Response · Authors · 2026-04-28
> **...**
>
> ## (2a) Improved encoder performance
> Prompted by the reviewer's comment, we have improved the performance of QA-based encoder by almost 10\%:
>  - cosine similarity of 0.53 (+9\%; previous: 0.44),
> -  binary accuracy of 0.74 (+4\%; previous 0.70).
>
> Specifically, the following minor changes have been implemented to achieve this performance improvement while leaving subsequent findings unchanged:
>  1. **TRAINING DATA**: We improved the *handling of compound nouns during lexical semantic extraction* for discovery concept-oriented features of the dataset questions and answers.
>     This also leads to decreasing the training examples $\mathcal{Q}$ by 8\% (from 693K to 634K).
>  2. **ENCODER'S HYPERPARAMETERS**: Reduce the *probability of drop-out layers* of the encoder model, from 0.5 to 0.1. Since the number of model's parameters is relatively low (55-71M), the previous value was too aggressive.
>
> ## (2b) QA-based insights on another language model
> In addition, the revised version now **also includes an evaluation of the QA-based setup using GPT-2** (Section 6), a *smaller language model* (355M parameters) compared to the 8B-parameter llama.
> - The encoder trained on GPT-2’s neural embeddings achieves performance *comparable to llama’s training results*, and it extracts an average of six concepts both before the first token and after the last token generation.
> - This cross-model comparison *reinforces previous findings about latent representations*:
>     1. the loss of question-related concepts after text generation, and
>    2. GPT-2’s internal states reflecting its overreliance on in-context information.
>
> # (3) Concept codebook for open-domain tasks 📚
> The concept codebook can *be viewed as a vocabulary with no practical constraints* on the cardinality, type, or origin of its symbols, enabled by the properties of VSAs. Analogous to a conventional vocabulary, it may include all possible syntactic and semantic features, only task-relevant features, or even mirror an LLM’s output vocabulary (inheriting, however, the limitations of token-based analysis).
>
> Importantly, **the probe’s performance does not depend on how the concept codebook is constructed, since it functions solely as a list of possible input features.** Its effectiveness instead depends on the type of latent features being targeted, as is generally the case for interpretability probes: if the target features are not encoded in the representational space, the resulting analysis will be uninformative.
> On the other hand, the encoder/probe cannot extract latent features that it has never encountered, consequently, it must be exposed
> to a few examples of the latent features being targeted, whether drawn from in-domain or out-of-domain distributions.
>
> Accordingly, **the most critical aspect of our methodology is the construction of input representations (VSA encodings), rather than the definition of the concept set.**
> In this work, we empirically show that *VSA-based input representations can be built using two strategies*:
> 1. a predefined structure (analogy-style inputs; Section 5), or
> 2. a dynamic set of in-context concepts derived from lexical semantics (Section 6).
>
> ## Discussion regarding generalization
> The manuscript already discusses the *generalization* of input representations (Appendix I.1) and their applicability to downstream tasks (Appendix I.2), including a formulation for an open-domain setting that constructs inputs using conventional NLP tools such as POS tagging and Semantic Role Labeling (SRL).
> Further, the *QA setting (Section 6) can further be seen as open-domain*, as both the codebook and input representations are automatically constructed from open-domain lexical resources.
>
>  In response to the reviewer’s comment, *we further extend this discussion to address cases where domain-relevant concepts are not present in the codebook*, now Appendix I.3.
> **While we have improved references to these appendices in the main text, *incorporating their full content directly would have disrupted the readability and flow of the manuscript.***

---

### Author Response · Authors · 2026-07-20
**Revised version and authors' comments**

We sincerely appreciate the constructive feedback from all reviewers.
The reviewers recognized the *novelty* and *significance of our approach*, which combines Vector Symbolic Architectures (VSAs) with *neural probing for LLM interpretability* (eTLR, ToBn), providing a *novel alternative to existing supervised probing methods* (VyLY). They also appreciated the *comprehensive experimental evaluation*, which includes multiple LLMs of varying sizes and architectures (eTLR, VyLY) and covers both analogy-style completion and question-answering text generation tasks (VyLY). Finally, the reviewers found the manuscript to be *well organized* (VyLY) and *well written* (ToBn).

The second revised version of our manuscript (with changes highlighted in blue and those from the first revision in green) addresses the main  reviewers' concerns, including:

- **Comparison with SAEs** (eTLR): We now include an *experimental comparison* between our VSA-based results and those derived from Sparse Autoencoders (SAEs, Section 5.4).

- **Causal validation** (VyLY, ToBn):
While intervention-based analysis falls outside our primary scope and is not supported by our current unidirectional design, this transformation serves as a key building block toward active LLM steering. We explicitly address this constraint in both the Introduction (Section 1) and the Limitations paragraph of Section 7.
    As an extension, the revised manuscript now includes a *steering-based use case* in Appendix I.4 that leverages a bidirectional transformation to enable an intervention-based analysis of language generation. Building on these promising results, we are developing a comprehensive VSA-based steering methodology, which we will present in a future standalone publication.

- **Claim on concept-driven patterns** (VyLY): We now *better present our findings as experimental observations* of the concept-based compositionality of LLM inference, rather than as evidence suggesting a causal mechanism.
    Sections 5 and 6 have been revised to provide a clearer introduction to the two core concept-focused dimensions investigated in this work: target concept retrieval (Figure 4) and compositional concept extraction (Table 1, 3 and 4).

- **Deeper investigation for the QA-based setup** (eTLR; Section 6).
        Firstly, we have *improved the performance of the QA-based encoder* by almost 10\%, keeping our subsequent findings unaffected.
        The training performance of the encoders is also compared with a baseline null model.
        In addition, Section 6 now includes *experimental results with a further and diverse language model* (GPT-2).